# L-Eval: Instituting Standardized Evaluation for Long Context Language Models

## Abstract

Recently, there has been growing interest in extending the context length of large language models (LLMs), aiming to effectively process long inputs of one turn or conversations with more extensive histories. While proprietary models such as GPT-4 and Claude can largely preserve the reasoning ability in an extended context, open-source models are still progressing through the early stages of development. To bridge this gap, we propose L-Eval to institute a more standardized evaluation for long context language models (LCLMs) addressing two key aspects: dataset construction and evaluation metrics. On the one hand, we build a new evaluation suite containing 20 sub-tasks, 508 long documents, and over 2,000 human-labeled query-response pairs encompassing diverse question styles, domains, and input length (3k~200k tokens). On the other hand, we investigate the effectiveness in evalution metrics for LCLMs. Results show that popular n-gram matching metrics generally can not correlate well with human judgment, and thus we strongly advocate for length-instruction-enhanced (LIE) evaluation and employing LLM judges. We conducted a comprehensive study of 4 popular commercial LLMs and 12 open-source counterparts using the L-Eval benchmark. Our empirical findings offer useful insights into the study of LCLMs and lay the groundwork for the development of more principled evaluation of these models.

## 1 Introduction

Currently, a significant amount of effort is being dedicated to research on extending the context length of large language models. Popular solutions mainly involve further pretraining or finetuning standard models on longer inputs using more efficient architectures (Ding et al., 2023; Dao et al., 2022; Liang et al., 2023; Mohtashami & Jaggi, 2023; Li et al., 2023b), as well as scaled positional embedding (Su et al., 2022; Sun et al., 2022; LocalLLaMA, 2023b; Qin et al., 2023).

There are extensive multi-task benchmarks (Hendrycks et al., 2021a; Suzgun et al., 2022) for language models with short prompts, yet a high-quality one in long context modeling has not yet been established, presenting an opportunity for further development in this area. Meanwhile, almost all previous long-sequence text generation benchmarks relied primarily on n-gram matching metrics (Zhang et al., 2023; Shaham et al., 2022), such as ROUGE (Lin, 2004). Whether these commonly used metrics correlate well with human judgment when testing LCLMs in a zero-shot setting remains a question. Furthermore, the open-source community has released a considerable number of language models with 16k, or 32k context length (Li et al., 2023a; Du et al., 2022). A comprehensive comparative study of these models can be of great value.

To address these issues, we propose *L-Eval* to call for a more standardized evaluation of long context language models. For dataset construction, L-Eval has 20 sub-tasks, 4 sub-tasks are annotated from scratch (§3.1), 4 sub-tasks are re-annotated from the public datasets (§3.2), and the remaining 12 sub-tasks are manually cleaned from previous long sequence datasets. We divide these tasks in L-Eval into two groups: closed-ended tasks and open-ended tasks. The closed-ended group primarily tests the reasoning and understanding ability regarding a longer context, and the open-ended group consists of more summarization tasks that require aggregation of long document information. In the design of L-Eval, we prioritize diversity and quality over quantity, ensuring correctness by manually validating all samples after data collection (§3.3). Our data diversity, indicative in question styles, domain selection, and input lengths, is detailed in Table 1.

In addition, the development of suitable evaluation metrics for LCLMs on open-ended tasks where multiple outputs are acceptable is crucial, yet challenging. In this work, we study the limitations of traditional metrics based on lexical matching. We demonstrate that these metrics often fail to correlate with human evaluation results. Our further experiments suggest that LLM judges (Li et al., 2023c; Zheng et al., 2023) provide superior accuracy in the evaluation of open-ended tasks. §4 explains how we set a short-context LLM judge in a long-context evaluion setting. Considering the influence of generation length on performance and in order to avoid drawing misleading conclusions, we propose the Length-Instruction-Enhanced (LIE) evaluation technique for all reference-based metrics, including those employing an LLM judger. The empirical results demonstrate a substantial improvement brought by LIE evaluation in the Kendall-Tau correlation coefficient ($\tau$) with human judgments (Figure 2), for all automatic metrics.

We also conducted a comprehensive study with 16 different LLMs (§5.1) in L-Eval. Some of our key findings are summarized below: (1) There is still a significant gap between open-source LCLMs and commercial models, for both closed-ended tasks (Table 3) and open-ended tasks evaluated by LLMs and human (Table 4, 5). However, this gap is not accurately reflected by n-gram metrics. (2) While current efforts on open-source LCLMs improve performance on closed-ended tasks, they significantly fall short on open-ended tasks. This is largely due to the models' misunderstanding of instructions as the input context length increases. (3) Experiments on GPT-3.5-Turbo with both dense and sparse retrievers show that end-to-end full-context models outperform traditional retrieval-based systems. (4) Training-free scaled positional embeddings can enhance the retrieval capability of LLMs over longer input, while it may adversely affect their reasoning ability.

More interesting conclusions can be found in §5.2 and §A.3. We hope *L-Eval* and our findings contribute to a deeper understanding of current LCLM research and the further development of models and evaluation metrics. All data, code and inference results in L-Eval will be released.

## 2 RELATED WORK

### 2.1 LONG CONTEXT LANGUAGE MODELS

Feeding long context leads to bottlenecks in language model training and inference due to computational resources. Some community efforts focus on developing efficient attention mechanisms to build efficient language models (Sun et al., 2023; Ding et al., 2023; Li et al., 2023b; Fu et al., 2023; Peng et al., 2023a). In addition to optimizing the attention mechanism, some works (Bulatov et al., 2023; Dai et al., 2019; Mohtashami & Jaggi, 2023) focus on chunking the input to model both the current text in the chunk and the previous context states, effectively extending the length of context processing. Besides the efficiency challenge, the scalability of positional embedding is also crucial. ALiBi (Press et al., 2022), and xPos (Sun et al., 2022) emphasize the significance of local context to enhance the language model's ability to perform extrapolation. Moreover, position interpolation (PI) (Chen et al., 2023) and NTK-aware (LocalLLaMA, 2023b;a) are the most popular approaches based on RoPE (Su et al., 2022) to efficiently and effectively extend the context length. However, these works mainly validated their methods with perplexity (PPL) (Sun et al., 2021; LocalLLaMA, 2023b), and there has not been systematic validation on practical tasks.

### 2.2 LONG SEQUENCES BENCHMARKS

Tay et al. (2020) introduce the Long Range Arena (LRA), a benchmark encompassing five distinct classification tasks. CAB (Zhang et al., 2023) is another benchmark for different efficient attention designs by comparing both efficiency and accuracy. In language domain, previous work on LCLMs tends to report PPL to evaluate language models (Su et al., 2022; Peng et al., 2023b) on longer context. However, PPL may not usually correlate with the actual performance (Sun et al., 2021). ZeroScrolls (Shaham et al., 2022; 2023) and LongBench (Bai et al., 2023) are concurrent long context evaluation suites. L-Eval differs from them in 3 aspects: (1) Manually selected samples. Testing samples are automatically filtered by their benchmarks, while those for L-Eval are manually filtered. (2) Standardized metrics. We are the first to investigate the correlations between traditional lexical metrics and recently proposed LLM metrics with human judgment on Long context settings. L-Eval no longer mainly relies on N-gram metrics. (3) More closed-ended tasks. Due to fairness issues in open-ended tasks. L-Eval has more closed-ended tasks reflecting unbiased results.

## 3 TOWARDS HIGH-QUALITY AND DIVERSE LONG CONTEXT DATASETS

In this section, we highlight some key procedures in L-Eval data construction. Concretely, we show the annotation, re-annotation, and manual filtering pipeline and the statistics of L-Eval. Please refer to Appendix B for the complete annotation details and examples.

### 3.1 DATA ANNOTATION FROM SCRATCH

There are 4 datasets annotated from scratch in L-Eval: Coursera, SFcition, CodeU, and LongFQA. The original resources are videos from Coursera, previous open-source datasets, source code from famous Python libraries, and public earning call transcripts, respectively.

**Coursera**   This dataset originates from the Coursera website.[1] To reduce the difficulty of annotation, we choose four public courses related to big data and machine learning (§B.4). The input long document is the subtitles of the videos. Questions and the ground truth answers are labeled by the authors. The instruction style of Coursera takes the format of multiple choice. In order to increase the difficulty of the task, we have set **multiple correct options**. To the best of our knowledge, this is the first multi-choice dataset with multiple correct answers and it is more challenging than single-option questions (Table 3).

**SFcition**   We annotate this sub-task to test the loyalty of the LCLM to the input context. We argue that in LCLMs, contextual knowledge (stored in long input) is more crucial than parametric knowledge (gained during pretraining). Practically, many long documents are private and can never be seen during pretraining. LLMs should follow the contextual knowledge instead of parametric knowledge in long context settings. To simulate this scenario, we annotate a science fiction dataset consisting of True or False questions. Most of the answers to these questions contradict real-world principles and do not comply with actual physical laws (§B.5). We find that Turbo-16k struggles on this task, which tends to answer questions relying on parametric knowledge (Table 3).

**CodeU**   As a code understanding dataset, it requires LLM to infer the output of a lengthy Python program. We mainly use source code from Numpy[2] and construct a string processing codebase. To prevent LLMs from answering the question based on their parametric knowledge, we replace the original function name. LLMs should first locate where the function is called and determine which functions are invoked. CodeU is the most challenging task in L-Eval (§B.6).

**LongFQA**   We also notice that there is a lack of long context question answering datasets in the finance domain and we annotate the QA pairs based on public earning call transcripts from the *Investor Relations* section of 6 company websites. Please refer to §B.8 for details.

### 3.2 DATA RE-ANNOTATION FROM PUBLIC DATASETS

We re-annotate 5 publicly available datasets in L-Eval. **GSM(16-shot)** is derived from 100-grade school math problems in the GSM8k dataset (Cobbe et al., 2021). If the LCLM maintain its reasoning ablilty on longer context, ultizing more high-quality examples will a positive effect on solving math problems (Li et al., 2023b). We construct 16 in-context examples with lengthy Chain-of-Thought where 8 examples come from *chain-of-thought-hub*[3] and 8 examples are constructed by us. We experiment with the newly constructed examples and the accuracy of Turbo-16k-0613 rises from 79 (8-shot) to 84 (16-shot).

We inject come new synthesis instructions to test global context modeling into **QuALITY** (Pang et al., 2022), such as *"What can we infer from the longest sentence in this story?"* and *"How many words are there in the story?"*. Given that these types of questions may rarely occur in real-world conversations, their proportion in L-Eval is extremely small. The **Openreview** dataset contains papers collected from `openreview.net`. We ask the model to (1) write an Abstract section, (2) summarize the related work, and (3) finally give feedback including valuable suggestions and some questions for the authors. We select the paper with high-quality related work sections and helpful

---

[1] https://coursera.org/

[2] https://github.com/numpy/numpy

[3] https://github.com/FranxYao/chain-of-thought-hub

reviews written by human reviewers to form this test set.[4] Next, we use **SPACE** (Angelidis et al., 2021) to test the aspect-based review summarization task, and the instructions for the dataset are annotated by us. We adopt diverse instructions to prevent overfitting.

Previous work (Li et al., 2023a; Liu et al., 2023) has used retrieval tasks to test the ability of modeling long context dependency via retrieving something over lengthy context. L-Eval includes a popular first topic retrieval task **TopicRet** (Li et al., 2023a), formatted as: "*[topic-1] Chat History [instruction]*". However, as we can see from Figure 1, retrieving the first topic is too easy to distinguish the ability of different models. However, the task of retrieving the second and the third topics presents a significantly higher level of challenge. It is observed that nearly all open-source models struggle in task. So we enhance the task with second/third topic retrieval.

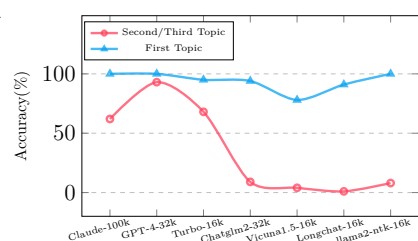

Figure 1: Test Accuracy (%) of different models with retrieving the first topic and retrieving the second/third topic.

### 3.3 DATA FILTERING AND CORRECTION

The remaining 12 tasks originates from existing datasets following previous evaluation suites (Zhang et al., 2023). However, L-Eval involves more human labor after data collection because we find the annotation quality of previous long sequence datasets fluctuates severely and there are many unanswerable questions that are unrelated to the context. These mistakes can hardly be corrected using the automatic preprocessing scripts in previous works. In L-Eval, all samples are manually filtered and corrected after data collection. Specifically, we use Claude-100k as our assistant to filter mistaken QAs and unanswerable questions. First, we input the lengthy document into Claude and request it to provide the answer and offer an explanation. If Claude produces an answer greatly mismatching the ground truth or states that we cannot deduce the answer from the context, we will either perform re-annotation or simply remove them.

### 3.4 STATISTICS

The statistics of L-Eval are shown in Table 1. The L-Eval contains various question styles such as multiple choice questions (TOFEL (Tseng et al., 2016), QuALITY, Coursera), true or false questions (SFiction), math problems (GSM), code understanding (CodeU), goal-oriented dialogues (Multi-Doc2Dial (Feng et al., 2021)), extractive QA (CUAD (Hendrycks et al., 2021b), NQ (Kwiatkowski et al., 2019)), abstractive QA (LongFQA, NarrativeQA (Kočiský et al., 2017), Qasper (Dasigi et al., 2021)), single document summarization (GovReport (Huang et al., 2021), BigPatent (Sharma et al., 2019), SummScreen (Chen et al., 2022), QMSum (Zhong et al., 2021)), multi-document summarization (Multi-News (Fabbri et al., 2019), SPACE (Angelidis et al., 2021)), research writing (Openreview) and so on. The long documents in L-Eval across many domains such as law, finance, academic papers, lectures, lengthy conversations, news, famous Python codebase, long-form novels, and meetings. The average input length in L-Eval ranges from 4k to 60k. The maximum sample in L-Eval contains nearly 200k tokens. This diversity represents real-world scenarios where different tasks may require different lengths of context and instructions. The length of reference in L-Eval also varies significantly across tasks.

## 4 TOWARDS STANDARDIZED LONG CONTEXT EVALUATION METRICS

In this section, we present various evaluation metrics for text generation, including exam evaluation for close-ended tasks and different levels of open-ended evaluation, most of which are reference-based metrics. We also conduct experiments to study the correlation between automated metrics and human scoring.

---

[4]Ethic statement: we discourage reviewers from using large models for reviews. Our goal is to assist authors in further improving their papers.

Table 1: This table presents the statistics of the L-Eval suite where **Question-style** indicates the type of task or the style of instruction in the dataset, **#Doc** refers to the number of long documents, and **#Instr** denotes the number of instructions provided for each long input. **Avg/Max len** signifies the average/maximum length of the document inputs. We tokenize the raw text with Llama2 tokenizer and report the number of tokens.

| Dataset | Question-style | Domain | Avg len | Max len | #Instr | #Doc |
|---------|---------------|--------|---------|---------|--------|------|
| *Closed - Ended Tasks* | | | | | | |
| TOEFL | Multiple choice | English test | 3,907 | 4,171 | 269 | 15 |
| GSM(16-shot)[†] | Solving math problems | In-context examples | 5,557 | 5,638 | 100 | 100 |
| QuALITY [†] | Multiple choice | Gutenberg | 7,169 | 8,560 | 202 | 15 |
| Coursera* | Multiple choice | Advanced courses | 9,075 | 17,185 | 172 | 15 |
| TopicRet[†] | Retriving topics | Conversation | 12,506 | 15,916 | 150 | 50 |
| SFcition* | True or False Questions | Scientific fictions | 16,381 | 26,918 | 64 | 7 |
| CodeU* | Deducing program outputs | Python Codebase | 31,575 | 36,509 | 90 | 90 |
| *Open - Ended Tasks* | | | | | | |
| MultiDoc2Dial | Goal-oriented dialogues | Grounded documents | 3,905 | 7888 | 136 | 20 |
| Qasper | QA on papers | NLP papers | 5,019 | 6,547 | 160 | 20 |
| LongFQA* | QA on earning call | Finance | 6,032 | 7824 | 52 | 6 |
| NQ | QA from Google Search | Wikipedia | 23,698 | 47,726 | 104 | 20 |
| CUAD | Extracting key information | Law | 30,966 | 68,625 | 130 | 20 |
| NarrativeQA | QA on narratives | Gutenberg | 62,335 | 210,541 | 182 | 20 |
| Multi-News | Multi-doc Summarization | Multiple News articles | 7,320 | 19,278 | 11 | 11 |
| GovReport | Single-doc Summarization | Government reports | 7,495 | 27,128 | 13 | 13 |
| BigPatent | Single-doc Summarization | Lengthy patents | 7,718 | 12,867 | 13 | 13 |
| SummScreen | Transcripts Summarization | TV series transcripts | 10,688 | 14,544 | 13 | 13 |
| Openreview[†] | Paper writing & reviewing | Papers from Openreview | 11,170 | 33,303 | 60 | 20 |
| QMSum | Query-based summarization | Meeting transcripts | 16,692 | 33,310 | 156 | 20 |
| SPACE[†] | Aspect-based summarization | Reviews on Hotels | 19,978 | 22,158 | 120 | 20 |

**Exam evaluation** This is designed for closed-ended tasks, i.e., multiple-choice questions. The evaluation metric used for these tasks follows the exact match format (accuracy %), similar to grading exam papers. Each question's score is calculated as 100 divided by the number of questions.

**Human evaluation** This is the most accurate evaluation for open-ended tasks. Despite that some works show GPT-4 can be coherent with human judgment, LLMs cannot replace human evaluation. We engage human evaluators to score the outputs on a scale of 1 to 5, which signifies from poor output to excellent output. To save human laboratories, we propose a subset used for the human evaluation which has 12 long documents with 85 open-ended questions (**85-question subset**).

**Large language model judges for evaluating LCLMs** In short context settings, evaluation using LLMs is the most accurate metric for automatically evaluating models on open-ended tasks (Zheng et al., 2023; Li et al., 2023c; Dubois et al., 2023). These works assume the LLM evaluator is a "super model", but this assumption does not hold in long context settings because it's impossible to feed the entire lengthy inputs into LLMs like GPT-4. Unlike short context evaluation, GPT-4 is unable to infer the ground truth answer itself. Consequently, evaluation results mainly depend on the reference answer and user questions. In L-Eval, we take the pair-wise battle format and we select Turbo-16k-0613 as the base model and report the *win-rate vs. Turbo-16k-0613* % which means how many samples can beat Turbo-16k. We study two LLM judges: GPT-4 and GPT-3.5 in the experiment section. LLM evaluators have been reported to favor more detailed and lengthy answers (Zheng et al., 2023). This bias becomes more pronounced in long context settings as the invisible input makes it difficult for the judge to accurately determine the correctness of specific details and information. Therefore, the judgment model must bear in mind that details not corroborated by the reference answers should not be considered beneficial. We enhance the judgment prompt with: *Additional details or information that are not mentioned in the reference answer cannot be considered as advantages and do not let them sway your judgment.* If you only want to evaluate a portion of the tasks in L-Eval, we recommend using LLM judges. Verifying the 1000+ open-ended questions via GPT-4 is unaffordable.[5] Thus we manually split a subset for GPT-4 evaluation consisting of 17 diverse long documents with 96 open-ended questions (**96-question subset**).[6]

---

[5]Testing the 4 datasets in Table 2 needs about $100!

[6]Evaluating outputs from the 96-question subset with GPT-4 only needs about $5.

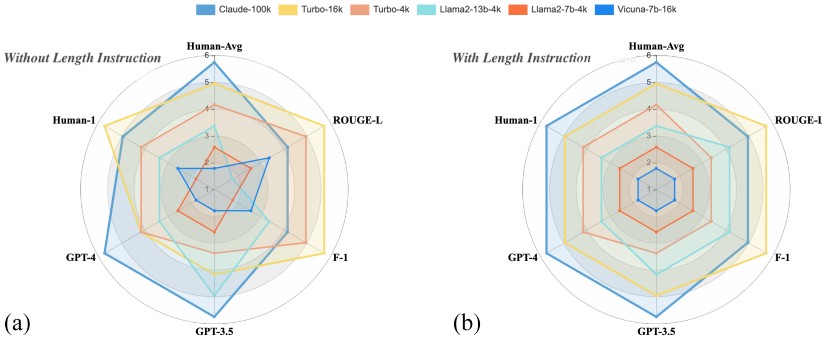

Figure 3: The ranking of six models under various evaluation metrics (Human-avg, Human-1, GPT-4, GPT-3.5, R-L, and F-1) with or without length instruction. Human-avg represents the average score from human evaluation, and Human-1 signifies the score given by the first human annotator.

**N-gram matching evaluation** Considering that assessing all tasks is still expensive for human/LLM evaluators, L-Eval also takes into account n-gram metrics. N-gram metrics like ROUGE-L (R-L) and F-1 score are widely used in traditional datasets and they are also widely adopted in the text generation benchmarks via performing lexical matching. It is worth noting that n-gram matching metrics are very sensitive to the length of the ground truth, exhibiting a length bias. The related analysis is in the following §4.1.

## 4.1 LENGTH INSTRUCTION ENHANCED LONG CONTEXT EVALUATION

In preliminary experiments, we find that LLMs tend to generate very long responses bringing obstacles for the reference-based evaluation (see $\Delta$**L** Table 2). This length bias results in a significant influence on the n-gram metrics. For instance, Claude-100k only achieves a 9.84 F-1 score due to undesired output length.

In L-Eval, we argue that long context language models should further focus on more accurate content rather than accurate length. Practically, issues about undesired generation length can be easily solved by prompting the model. We first adopt **Length-Instruction-Enhanced**

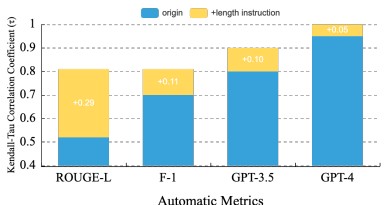

Figure 2: Kendall-Tau correlation coefficient of different automatic metrics with the average human score.

(LIE) evaluation in LLMs evaluation benchmarks which is simple but effective in overcoming the length bias, i.e., the number of words of ground truth is directly exposure to LCLMs. LIE evaluation in this work is implemented by injecting the model with the desired length into the original instruction (e.g., [Origin Instruction]: *Please summarize the opinions of the professor*. [Length Instruction]: *We need a 50-word summary*, where 50 is the number of words in the reference answer). The results of Claude-100k in Table 2 demonstrate a substantial improvement in terms of the F-1 score: there is a near **50**-point gap depending on whether or not the model generates with the expected length.

**Experimental validation** To validate the LIE evaluation, we then conduct a human evaluation on the 85-questions subset. We have 3 annotators to verify 7 models and calculate the Kendall-Tau correlation coefficient ($\tau$) between these metrics and the average human score. The main results are shown in Figure 2 (Blue bar) and experimental settings are in §A.2. Results indicate that all these automatic metrics (except GPT-4) **fail to correlate** to human judgment. Compared with N-gram metrics, LLM judges are more accurate and robust to output length. As we can see from Figure 2, the improvements brought by length instruction are marked with yellow, and after adding the length instructions, $\tau$ has been improved from 0.5 to 0.8 for ROUGE-L and $\tau$ of GPT-4 evaluator has even reached to 1. In Figure 3, we convert the score to rankings (the best one is 5 and the worst is 1) and show the score of 6 models evaluated with 6 different evaluation systems. Figure 3 (a) shows the results given by metrics without length instruction. These hexagons are often distorted because these metrics usually cannot achieve good correlation. When comparing the models enhanced with length instruction in (b), it is observed that the hexagons become more regular.

Table 2: Results on 2 open-ended summarization and 2 abstractive QA tasks. **GPT-4** means the win-rate with Turbo-16k using GPT-4 as the judge. **ΔL** means the difference of generated answer length with ground truth length. The best results are underlined. Results in red mean decoding in a desired length makes a big difference in performance.

| Model | SPACE | | | QMSum | | | NQ | | | NrtvQA | | |
|---|---|---|---|---|---|---|---|---|---|---|---|---|
| | R-L | GPT-4 | ΔL | R-L | GPT-4 | ΔL | F-1 | GPT-4 | ΔL | F-1 | GPT-4 | ΔL |
| Claude-100k | 15.43 | 45.65 | **165** | 14.04 | 58.77 | **183** | 9.84 | 56.19 | **135** | 10.39 | 68.96 | **127** |
| + Length Instruction | 18.61 | 61.40 | 27 | 18.13 | 58.89 | 22 | 57.76 | 51.00 | 1 | 19.09 | 57.77 | 0 |
| Chatglm2-32k | 17.56 | 24.13 | -23 | 20.06 | 38.84 | **287** | 31.45 | 33.71 | 3 | 12.24 | 34.67 | 74 |
| + Length Instruction | 16.61 | 17.11 | 11 | 20.83 | 33.75 | 9 | 37.94 | 33.71 | -1 | 14.00 | 34.52 | -2 |
| Longchat-7b-16k | 15.10 | 15.61 | **120** | 9.31 | 25.56 | 40 | 8.83 | 32.33 | **105** | 8.36 | 31.80 | 83 |
| + Length Instruction | 17.06 | 36.23 | -3 | 13.21 | 30.20 | 70 | 20.21 | 35.00 | 37 | 15.17 | 43.38 | 40 |
| Llama2-13b-chat | 16.83 | 32.46 | **102** | 14.72 | 30.79 | **116** | 8.29 | 38.99 | 90 | 7.20 | 30.69 | **130** |
| + Length Instruction | 19.23 | 43.15 | -7 | 19.65 | 34.82 | -1 | 35.43 | 41.07 | 6 | 13.48 | 45.07 | 14 |

## 5 BENCHMARKING LLMs WITH L-EVAL

In this section, we list our 16 baseline models and the results on both open-ended and closed-ended tasks. Generally, there are considerable gaps between open-source models and commercial models. A detailed description of baseline models can be found in §A.1. The prompt templates for each task are available in §B. We run all the experiments using FlashAttention (Dao et al., 2022) on a single NVIDIA A800 GPU. The document input is truncated from the right.

### 5.1 BASELINES

**Commercial Models** (1) Claude-100k developed by Anthropic, (2) GPT-4-32k, OpenAI's most powerful long context model, (3) Turbo-4k-0613 and (4) Turbo-16k-0613 is the snapshot of GPT-3.5 from June 13th 2023 which can handle up to 4k/16k input tokens.

**Open-source Models** (5) Llama1 (Touvron et al., 2023a), a widely used open-source model developed by Meta AI with a 2k pre-training length, (6) Vicuna1.3 (Chiang et al., 2023), tuned on shareGPT based on Llama1, (7) Longchat-16k, the long context version of Vicuna1.3 using PI, (8) Llama2, the next version of Llama with 4k pre-training context, (9) Llama2-chat, a finetuned version for dialogue usage, (10) Llama2-NTK, extending the context length of Llama2-chat with NTK-aware RoPE, (11) Vicuna1.5-16k (Zheng et al., 2023), the long context version of Llama2 using PI & ShareGPT (12) Longchat1.5-32k, the 32k context version of Llama2 using PI & ShareGPT. (13) Chatglm2-8k, the second version of the Chatglm (Du et al., 2022), (14) Chatglm2-32k, the 32k context length version, (15) XGen-8k-inst (Nijkamp et al., 2023), an 8k context models developed by salesforce (16) MPT-7B-StoryWriter-65k, based on MPT-7B and ALiBi with a context length of 65k tokens on a subset of Books3 dataset.

**Retriever** We implement the dense retriever with the OpenAI AdaEmbedding as the dense retriever and BM25 as the sparse retriever to extract 4 pieces of most related 1k-chunked documents, which are further provided as the context to answer questions.

### 5.2 MAIN RESULTS

The performance of LCLMs on closed-ended tasks is shown Table 3. As for open-ended tasks, we test the 96-question subset (Table 4) with GPT-4 evaluation. Results from n-gram metrics on all test sets and the rankings of LLMs can be found in §A.3. From the main results, we have the following observations. GPT-4-32k clearly outperforms all other models by a very significant margin, establishing SOTA in L-Eval closed-ended tasks. There is still a near **20**-points gap between the best open-source 16k models and Turbo-16k. As for open-ended tasks, since the input texts are generally longer and a global understanding of the context is required, Claude-100k, with the longest context length, surpasses all baseline models including GPT-4-32k. Although results of n-gram metrics indicate that open-source LCLMs have achieved performance close to GPT-Turbo on open-ended tasks, the evaluation outcomes from both LLM (Table 4) and human judges (Table 5) reveal that there is still a significant gap between them. Moreover, retrieval-based methods based on Turbo-4k fall short in comparison to encoding the entire context (Turbo-16k), as certain tasks are difficult to address through simple retrieval.

Table 3: Exam evaluation results on **closed-ended tasks** for current LCLMs. **Ret.** indicates whether we use retrieve-based algorithms for the base model. **Tokens** denotes the maximum number of input tokens we feed into the model. ↓/↑ indicates a remarkable decrease/increase in performance, compared to using the original short context counterpart. * indicates the model is not further trained.

| Model | Ret. | Tokens | Coursera | GSM | QuALITY | TOEFL | CodeU | SFiction | Avg. |
|---|---|---|---|---|---|---|---|---|---|
| Claude1.3-100k | ✗ | 100k | 60.03 | 88.00 | 73.76 | 83.64 | 17.77 | 72.65 | 65.97 |
| GPT-4-32k | ✗ | 32k | **75.58** | **96.00** | **82.17** | **84.38** | **25.55** | **74.99** | **73.11** |
| Turbo-16k-0613 | ✗ | 16k | 63.51 | 84.00 | 61.38 | 78.43 | 12.22 | 64.84 | 60.73 |
| AdaEmb-Turbo-4k-0613 | ✓ | 4k | 61.77 | 23.00 | 58.91 | 76.95 | 6.66 | 71.09 | 49.73 |
| BM25-Turbo-4k-0613 | ✓ | 4k | 63.80 | 23.00 | 59.40 | 75.09 | 5.55 | 71.09 | 49.65 |
| *Truncating input tokens to the pretraining context length* | | | | | | | | | |
| Llama1-7b-2k (w/o SFT) | ✗ | 2k | 13.37 | 7.00 | 21.78 | 30.85 | 1.11 | 35.15 | 19.22 |
| Vicuna1.3-7b-2k | ✗ | 2k | 34.73 | 19.00 | 32.67 | 43.49 | 1.11 | 60.93 | 30.01 |
| Llama2-7b-4k (w/o SFT) | ✗ | 4k | 20.05 | 2.00 | 28.71 | 24.53 | 0.00 | 40.62 | 19.31 |
| Llama2-7b-chat | ✗ | 4k | 29.21 | 19.00 | 37.62 | 51.67 | 1.11 | 60.15 | 33.12 |
| Llama2-13b-chat | ✗ | 4k | 35.75 | **39.00** | **42.57** | **60.96** | 1.11 | 54.68 | **39.01** |
| Chatglm2-6b-8k | ✗ | 2k | **43.75** | 13.00 | 40.59 | 53.90 | 2.22 | 54.68 | 34.69 |
| XGen-7b-8k (2k-4k-8k) | ✗ | 2k | 26.59 | 3.00 | 35.15 | 44.23 | 1.11 | 48.43 | 26.41 |
| *Truncating input tokens to the further finetuning context length* | | | | | | | | | |
| Chatglm2-6b-32k | ✗ | 32k | **47.81** | 27.00↑ | 45.04 | 55.01 | 2.22 | 57.02 | 39.01↑ |
| Longchat1.5-7b-32k | ✗ | 32k | 32.99 | 18.00 | 37.62 | 39.77 | 3.33 | 57.02 | 31.45 |
| Longchat-7b-16k | ✗ | 16k | 29.74 | 10.00↓ | 33.66 | 47.95 | 3.33 | **64.84** | 31.58 |
| Vicuna1.5-7b-16k | ✗ | 16k | 38.66 | 19.00 | 39.60 | 55.39 | **5.55** | 60.15 | 36.39↑ |
| Llama2-7b-NTK* | ✗ | 16k | 32.71 | 19.00 | 33.16 | 52.78 | 0.00 | **64.84** | 33.74 |
| Longchat-13b-16k | ✗ | 16k | 31.39 | 15.00 | 40.59 | 55.39 | 2.22 | **64.84** | 34.90 |
| Vicuna1.5-13b-16k | ✗ | 16k | 40.69 | 36.00 | **53.96**↑ | **68.40**↑ | 0.00 | 61.71 | **43.46**↑ |
| Llama2-13b-NTK* | ✗ | 16k | 36.48 | 11.00↓ | 35.64 | 54.64 | 1.11 | 63.28 | 33.69 |
| Llama2-13b-NTK(Dyn)* | ✗ | 16k | 30.08 | **43.00** | 41.58 | 64.31 | 1.11 | 35.15 | 35.87 |
| Chatglm2-6b-8k | ✗ | 8k | 42.15 | 18.00 | 44.05 | 54.64 | 2.22 | 54.68 | 35.95 |
| XGen-7b-8k | ✗ | 8k | 29.06 | 16.00 | 33.66 | 42.37 | 3.33 | 41.40 | 27.63 |
| MPT-7b-65k | ✗ | 8k | 25.23 | 8.00 | 25.24 | 17.84 | 0.00 | 39.06 | 19.22 |

**Fine-tuning longer offers benefits for closed-ended tasks but falls short in open-ended tasks** In Table 3, for open-source models using scaled positional embedding, Longchat and Vicuan1.5-16k obviously outperform their original version Vicuna-2k and Llama2-chat. The results suggest that further tuning on longer input from a model with short pretraining context length does benefit long context modeling. However, according to Table 4, unlike results on closed-ended tasks, the best model Vicuna1.5-13b-16k only wins Turbo-16k by 34%, **8** points lower than its short version Llama2-13b. Llama2-13b-chat (Touvron et al., 2023a) is still the strongest

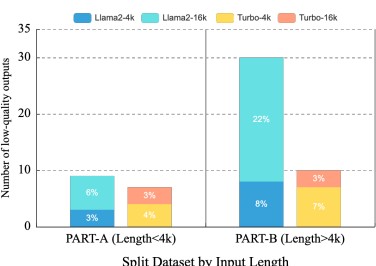

Figure 4: Number of invalid outputs from Llama2 and Turbo.

open-source baseline, indicating that current LCLMs simply based on scaled position embedding may not be enough for these challenging open generation tasks. Based on our human evaluation, we find that although scaled position embedding techniques such as NTK (LocalLLaMA, 2023b) or PI (Sun et al., 2022) effectively extend models' context length, the models tend to get lost when facing lengthy input tokens and are unable to follow the instruction. We classify these outputs as "invalid outputs". To investigate model performance on different context lengths, we split the 85-questions subset into 2 parts: PART-A contains samples with less than 4k tokens, and PART-B more than 4k tokens. We compare the number of invalid outputs from Llama2/Vicuna1.5-16k and Turbo/Turbo-16k in Figure 4. Results show that the number of invalid outputs from Turbo-16k remains a very small amount on both PART-A and B while the invalid outputs from Llama2-16k dramatically increase on samples with longer input. Thus, LCLMs are less capable of following instructions on open-ended tasks for long contexts, compared with closed-ended tasks, such as multiple choice. A possible reason is that the pertaining or SFT corpus is highly likely to contain many

Table 4: In comparing various models to Turbo-16k-0613 on **open-ended tasks**. We evaluate these models on the 96-question subset using GPT-4 and two subsets (85+96 questions) using GPT-3.5. We reduce the positional biases by swapping paired predictions, so the GPT-4 evaluator is used in 96×2 evaluation rounds, while the GPT3.5 evaluator is used in 181×2 rounds

| Model | Ret. | Tokens | GPT-4 | | | GPT-3.5 | | | R-L |
|---|---|---|---|---|---|---|---|---|---|
| | | | wins | ties | win-rate % | wins | ties | win-rate % | |
| Claude1.3-100k | ✗ | 100k | **96** | 42 | **60.94** | 189 | 34 | **58.68** | 28.22 |
| GPT-4-32k | ✗ | 32k | 76 | 56 | 54.16 | 171 | 50 | 56.32 | 36.18 |
| Turbo-16k-0613 | ✗ | 4k | 0 | 192 | 50.00 | 0 | 362 | 50.00 | 28.61 |
| Turbo-4k-0613 | ✗ | 4k | 38 | 69 | 39.83↓ | 109 | 61 | 41.39 | 26.90 |
| AdaEmb-Turbo-4k-0613 | ✓ | 4k | 61 | 56 | 46.84 | 123 | 77 | 45.36 | 26.09 |
| BM25-Turbo-4k-0613 | ✓ | 4k | 50 | 69 | 44.01 | 125 | 78 | 45.30 | 26.83 |
| *Truncating input tokens to the pretraining context length* | | | | | | | | | |
| Vicuna1.3-7b-2k | ✗ | 2k | 29 | 55 | 29.42 | 97 | 42 | 34.91 | 16.17 |
| Longchat-7b-16k | ✗ | 2k | 26 | 63 | 29.94 | 87 | 38 | 31.26 | 19.77 |
| Llama2-7b-chat | ✗ | 4k | 48 | 58 | 40.10 | 127 | 44 | 42.45 | 24.25 |
| Llama2-13b-chat | ✗ | 4k | **51** | 61 | **42.44** | **143** | 49 | **47.85** | 24.07 |
| *Truncating input tokens to the further finetuning context length* | | | | | | | | | |
| Chatglm2-6b-32k | ✗ | 32k | 28 | 60 | 30.20 | 53 | 65 | 24.63 | 22.04 |
| Longchat1.5-7b-32k | ✗ | 32k | **38** | 53 | 33.59 | 136 | 37 | 44.91 | 21.21 |
| Longchat-7b-16k | ✗ | 16k | 36 | 56 | 33.68↑ | 108 | 42 | 37.94 | 20.59 |
| Vicuna1.5-7b-16k | ✗ | 16k | 22 | 54 | 25.52↓ | 102 | 52 | 37.86 | 18.05 |
| Llama2-7b-NTK* | ✗ | 16k | 18 | 49 | 22.13 | 58 | 35 | 23.59 | 11.50 |
| Longchat-13b-16k | ✗ | 16k | 36 | 59 | **34.11** | **128** | 24 | 40.11 | 18.98 |
| Vicuna1.5-13b-16k | ✗ | 16k | 36 | 59 | **34.11**↓ | 116 | 43 | **40.92** | 19.69 |
| Llama2-13b-NTK* | ✗ | 16k | 31 | 52 | 29.68 | 91 | 44 | 34.55 | 15.63 |
| Llama2-13b-NTK(Dyn)* | ✗ | 16k | 23 | 48 | 24.47 | 55 | 64 | 26.60 | 11.62 |
| Chatglm2-6b-8k | ✗ | 8k | 18 | 64 | 26.04 | 86 | 54 | 32.84 | 18.19 |
| XGen-7b-8k | ✗ | 8k | 24 | 62 | 28.64 | 89 | 72 | 36.02 | 20.51 |

training samples with similar question styles. This strongly enhances their instruction-following ability on closed-ended tasks.

**Performance on retrieval tasks contradicts reasoning tasks**
The most popular NTK-ware positional embedding methods increase the base 10,000 in the vanilla RoPE to implement extrapolation without further fine-tuning. However, we find that the performance on topic retrieval tasks does not match the reasoning capability over lengthy context. As can be seen from Figure 5, when we increase the base from 20,000 to 160,000, there is a continuous improvement on topic retrieval. However, performance on math reasoning tasks with lengthy examples exhibits a completely opposite trend, indicating that it is challenging for the model to maintain its reasoning abilities when increasing the base. In contrast, the performance on retrieval tasks seems to remain unaffected after the base reaches 60,000.

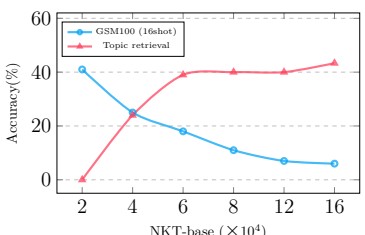

Figure 5: Test retrieval ability and reasoning ability with NTK base.

We have further analysis in §A.3, including full results of n-grams metrics on open-ended tasks, the rankings of current LLMs, NTK-aware positional embedding and retrieval-based systems.

## 6 CONCLUSION

In conclusion, the much-needed rigorous benchmark L-Eval introduced in this work provides a comprehensive suite of tasks and evaluation metrics to assess the capabilities of long context language models. We tested most of open-source LCLMs and experiments demonstrate promising gains from extending context length and gaps compared to commercial models. Our analysis using L-Eval offers valuable insights into the current state and limitations of LCLMs. We believe that with its focus on practical, long-form documents across domains, L-Eval can serve as a challenging testbed to drive advances in modeling longer contexts.

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

# A APPENDIX

## A.1 BASELINE MODELS IN L-EVAL

**Commercial Models**

- Claude-100k is developed by Anthropic[7] and targets understanding extremely long documents and answering related questions. It has the longest context length among all the LLMs.
- GPT-4-32k is developed by OpenAI[8]. It is the long context version of GPT-4 maintaining very strong reasoning ability over 32k context length but also the most expensive model.
- Turbo-4k-0613 is the snapshot of GPT-3.5[9] from June 13th 2023 which can handle up to 4k input tokens. Turbo-16k-0613 is the released long context version of Turbo-4k-0613.

**Open-source Models**

- Llama1 (Touvron et al., 2023a)[10] is a widely used open-source model developed by Meta AI with a 2k pre-training context length. The first version of Llama did not release a chatbot-based model.
- Vicuna1.3 (Chiang et al., 2023) [11] is a chatbot fine-tuned from Llama1 on shareGPT.
- Longchat-16k (Li et al., 2023a) [12] is the long context version of Vicuna. It uses positional interpolation to adapt 16k context. Concretely, they further fine-tune Llama1 on lengthy dialogues (16k tokens) from shareGPT.
- Llama2 (Touvron et al., 2023b) is the second version of Llama recently released by Meta AI. The updated version has 4k pretraining context with more powerful long context understanding capabilities.
- Llama2-chat (Touvron et al., 2023b) is a chatbot based on Llama2 released together with Llama2. Please notice that if we do not follow the pre-defined input format, i.e., ignore the special tokens, there will be a significant degradation in performance.
- Llama2-NTK-chat (LocalLLaMA, 2023b) is the long context version of Llama2-chat. It uses NTK-aware positional embedding. If we want to extend the model context window to $t$ (we call $t$ as a scale-up factor) times its original pertaining size, we just need to increase the original base=10,000 of RoPE ($\theta_n = 10000^{-2n/d}$) to $10{,}000 \times t^{\frac{d}{d-2}}$ where $d$ is the head dimension in Transformer. In our experiments, this theory does not hold in practical tasks (see section §A.3), which means the model still tends to generate random tokens when setting $t = 4$ on 16k context length. We set $t = 8$ in experiments.
- Llama2-NTK-chat (Dyn) (LocalLLaMA, 2023a) is the dynamic version of Llama2-NTK-chat. The only difference is that the scale-up factor $t$ in dynamic NTK depends on the current input length $L$ and the pertaining length $l$, i.e., $t = \frac{L}{l}$
- Vicuna1.5-16k uses Llama2 as the base model and performs further finetuning on concatenated 16k tokens lengthy dialogues from shareGPT. This model is based on positional interpolation which helps the training process converge fast.
- LongChat1.5-32k is the 32k version of Vicuna1.5-16k.
- Chatglm2-8k (Du et al., 2022)[13] is the second version of the open-source bilingual chat model Chatglm. The context length of the base model is further pretrained with 32k context window and finetuned on dialogue data with 8k context window.
- Chatglm2-32k is the long context version of Chatglm2 using positional interpolation.

---

[7]https://www.anthropic.com/index/100k-context-windows
[8]https://platform.openai.com/docs/models/gpt-4
[9]https://platform.openai.com/docs/models/gpt-3-5
[10]https://github.com/facebookresearch/llama
[11]https://github.com/lm-sys/FastChat.git
[12]https://github.com/DachengLi1/LongChat
[13]https://github.com/THUDM/ChatGLM2-6B

| Model | #Level-1 | #Level-2 | #Level-3 | #Level-4 | #Level-5 | Human-Avg | GPT-4 | GPT-3.5 | F-1 | R-L |
|---|---|---|---|---|---|---|---|---|---|---|
| *Length-instruction-enhanced evaluation reulsts* | | | | | | | | | | |
| llama2-7b-chat | 53 | 38 | 74 | 46 | 44 | 2.96 | 38.52 | 42.37 | 24.26 | 28.48 |
| llama2-13b-chat | 41 | 37 | 68 | 59 | 50 | 3.15 | 40.00 | 48.07 | 26.10 | 30.90 |
| Turbo-4k-0613 | 43 | 29 | 51 | 72 | 60 | 3.30 | 42.05 | 43.75 | 26.05 | 30.75 |
| Claude-100k | 14 | 15 | 37 | 69 | **120** | **4.04** | **60.88** | **63.75** | 26.39 | 31.57 |
| turbo-16k-0613 | 37 | 12 | 43 | **90** | 73 | 3.58 | 50.00 | 50.00 | **27.99** | **32.93** |
| Vicuan-7b-16k | 125 | 26 | 45 | 43 | 16 | 2.21 | 23.23 | 35.09 | 16.25 | 19.40 |
| longchat-7b-16k | 113 | 29 | 61 | 32 | 20 | 2.28 | 23.82 | 37.57 | 17.12 | 20.81 |
| *Original evaluation results* | | | | | | | | | | |
| llama2-7b-chat | 136 | 49 | 47 | 15 | 8 | 1.86 | 32.35 | 42.40 | 14.29 | 17.72 |
| llama2-13b-chat | 92 | 50 | 64 | 38 | 11 | 2.31 | 35.00 | 55.76 | 13.62 | 18.10 |
| Turbo-4k-0613 | 66 | 38 | 60 | 40 | 51 | 2.89 | 50.00 | 44.06 | 20.06 | 24.88 |
| Claude-100k | 27 | 52 | 81 | 66 | 29 | **3.08** | **53.23** | **76.68** | 15.31 | 19.59 |
| turbo-16k-0613 | 42 | 40 | 78 | 64 | 31 | 3.00 | 50.00 | 50.00 | 20.60 | **25.96** |
| Vicuan-7b-16k | 138 | 49 | 46 | 14 | 8 | 1.84 | 23.23 | 38.27 | 14.69 | 17.90 |
| longchat-7b-16k | 156 | 40 | 36 | 18 | 5 | 1.72 | 22.05 | 35.76 | 13.25 | 15.73 |

Table 5: Human evaluation results and results from other automatic metrics where **#Level-N** denotes the number of outputs (the sum from all annotators) in Level-N on the 85-question subset. Texts colored with red mean very unsatisfactory results.

- XGen-8k-inst[14] developed by salesforce follows a multi-stage pertaining procedure. They first train the model with 2k context length and progressively increase the pertaining length to 4k, finally reaching 8k.

- MPT-7B-StoryWriter-65k[15] is designed to handle super-long context lengths. It was tuned on MPT-7B with a context length of 65k tokens on a subset of Books3 dataset.

## A.2 HUMAN EVALUATION

Evaluating long-sequence, open-ended tasks remains a challenge. As previously discussed, almost all metrics, including the highly accurate automatic metric, the GPT-4 evaluator, exhibit bias. Consequently, human evaluation may be the most equitable metric to assess these models. In this section, we detail the human evaluation procedure conducted on seven baseline models using an 85-question subset. Our goal is to examine the correlation between human judgement and automatic metrics. Additionally, we aim to evaluate the performance of the length-instruction-enhanced evaluation method proposed in this paper.

**Experimental setup** We evaluate seven models, comprising three commercial and four open-source models: (1) Claude-100k, (2) turbo-16k-0613, (3) Turbo-4k-0613, (4) Vicuna1.5-7b-16k (Llama2), (5) Longchat-7b-16k (Llama1), (6) Llama2-7b-chat, and (7) Llama2-13b-chat. These models are tested on an 85-question subset from L-Eval open-ended tasks. Each sample is scored by three annotators, all of whom are Ph.D. students researching long context language models. We calculate the average score to obtain the final human evelation results. To determine if the ranking produced by these automatic metrics correlates with the ranking provided by the annotators, we use the Kendall-Tau correlation coefficient. We allow each model to generate outputs twice: first in the original mode without any length instruction, and then with the given length instructions. To minimize variance, we use greedy search as the decoding algorithm. The model outputs are ranked on a five-level scale:

- Level-1 (worst): The response is totally unhelpful to answer the question.

- Level-2: The output generally deviates from the original question, but some information is useful to solve the problem.

- Level-3: The response is partially correct, but the generated answer may contain some errors or omit key information.

---

[14]https://github.com/salesforce/xgen
[15]https://www.mosaicml.com/blog/mpt-7b

- Level-4: Most of the response is correct, but there may be minor issues such as being overly long (which cannot be considered a flaw if it is a reasonable explanation), or it might omit some information, but this does not affect the overall meaning.
- Level-5 (best): The output is close-to-human or even better.

**Human evaluation results** The results of our human evaluation are presented in Table 5. As can be seen, despite being fine-tuned on longer contexts, open-source long context models still struggle with very long input sequences during inference. When fed with numerous input tokens, the number of Level-1 outputs from open-source LCLMs significantly increases, while the LLMs with only a 4k context length can maintain their generation quality at a partially correct level, albeit without achieving high scores. It's also observable that the N-gram metrics F-1 and ROUGE generally do not correlate with the human evaluation results. Given the impracticality of testing a large number of samples using LLMs due to high costs and inefficiency, we also urge for more advanced metrics. We will release our human assessment to aid research on these metrics.

## A.3 ANALYSIS

| Model | Ret. | Tokens | Fin. | Contract | Multidoc | Nrtv | NQ | SCI | Avg. |
|---|---|---|---|---|---|---|---|---|---|
| Turbo-16k-0613 | ✗ | 16k | **45.36** | 24.87 | 31.45 | 18.20 | 45.90 | 28.25 | 32.33 |
| AdaEmb-Turbo-0613 | ✓ | 4k | 39.69 | 24.09 | **35.62** | **18.59** | 49.66 | **33.36** | **33.50** |
| BM25-Turbo-0613 | ✓ | 4k | 40.79 | **26.10** | 35.17 | 16.32 | **53.73** | 25.83 | 32.99 |
| *Truncating input tokens to the pretraining context length* | | | | | | | | | |
| Llama2-7b-chat | ✗ | 4k | **40.06** | 23.00 | **27.28** | 13.48 | 28.11 | 25.95 | 26.31 |
| Llama2-13b-chat | ✗ | 4k | 38.07 | **23.14** | 26.14 | **16.76** | 35.43 | 27.46 | **27.83** |
| Vicuna1.3-7b | ✗ | 2k | 30.49 | 17.69 | 17.70 | 14.57 | 15.49 | 7.69 | 17.27 |
| Longchat-7b-16k | ✗ | 2k | 27.27 | 19.78 | 13.99 | 13.21 | 18.11 | 7.61 | 16.66 |
| Chatglm2-6b-8k | ✗ | 2k | 29.60 | 19.06 | 16.22 | 13.21 | 17.52 | 12.26 | 17.97 |
| XGen-7b-8k (2k-4k-8k) | ✗ | 2k | 34.43 | 21.28 | 21.59 | 14.97 | 29.58 | 14.12 | 22.66 |
| *Truncating input tokens to the further finetuning context length* | | | | | | | | | |
| Chatglm2-7b-32k | ✗ | 32k | 30.27 | **26.95** | **24.97** | 14.00 | 37.94 | 26.44 | **26.76** |
| Longchat1.5-7b-32k | ✗ | 32k | 36.06 | 18.16 | 14.96 | 11.79 | 24.92 | 12.09 | 19.66 |
| Longchat-7b-16k | ✗ | 16k | 38.37 | 26.78 | 8.31 | **15.17** | 20.21 | 9.74 | 19.76 |
| Vicuna1.5-7b-16k | ✗ | 16k | 39.31 | 18.04 | 18.44 | 8.19 | 19.39 | 21.80 | 20.86 |
| Longchat-13b-16k | ✗ | 16k | 37.85 | 21.11 | 12.18 | 14.76 | 22.75 | 14.95 | 20.60 |
| Vicuna1.5-13b-16k | ✗ | 16k | **45.57** | 18.16 | 15.88 | 15.03 | 37.13 | 23.40 | 25.86 |
| Llama2-13b-NTK* | ✗ | 16k | 30.99 | 15.88 | 13.61 | 6.89 | 11.13 | 15.58 | 15.67 |
| Llama2-13b-NTK(Dyn)* | ✗ | 16k | 39.99 | 18.59 | 25.49 | 13.09 | 14.51 | **26.90** | 23.09 |
| Longchat-13b-16k | ✗ | 8k | 36.94 | 16.70 | 10.77 | 7.55 | 14.14 | 9.91 | 16.00 |
| Chatglm2-6b-8k | ✗ | 8k | 33.17 | 15.76 | 13.76 | 7.02 | 3.50 | 6.36 | 13.26 |
| XGen-7b-8k | ✗ | 8k | 36.40 | 22.01 | 17.08 | 9.41 | 13.88 | 20.23 | 19.83 |
| MPT-7b-65k | ✗ | 8k | 10.01 | 6.24 | 3.95 | 1.77 | 0.77 | 1.68 | 4.06 |

Table 6: Performance of various models on open-ended QA datasets in terms of F1 score. For results tested with N-gram metrics, please note that the results may not be accurate when the performance of the models is very similar or there is a large difference in the granularity of the output.

**Results from n-gram metrics** Test all cases in open-ended tasks in L-Eval with GPT-4 is affordable. To give an overview of all the models on open-ended tasks, we test all models with n-gram metrics. As can be seen from the win rate from LLM judges (Table 4) and human evaluation (Table 5), there is still a significant margin between commercial LLMs and open-source LLMs. However, the margin is not clear enough based on n-gram metrics. Based on n-gram metrics, the open-source LCLMs also fail to beat their origin short-context model on truncated context. Overall, current open-source LCLMs generally excel more in conventional **summarization tasks** that involve instructions like "*Summarize this document*" compared with query-based summarization and

| Model | Tokens | paper_assistant | | | review_summ | | | meeting_summ | | | Avg |
|---|---|---|---|---|---|---|---|---|---|---|---|
| | | R-1 | R-2 | R-L | R-1 | R-2 | R-L | R-1 | R-2 | R-L | |
| Turbo-16k-0613 | 16k | 39.55 | 10.92 | 18.61 | **30.18** | **7.14** | 18.67 | 30.20 | 7.22 | 19.31 | 20.20 |
| AdaEmb-Turbo-0613 | 4k | 38.07 | 9.61 | 17.33 | 29.81 | 6.47 | **18.91** | **31.92** | 8.24 | **20.84** | 20.13 |
| BM25-Turbo-0613 | 4k | **41.59** | **13.39** | **21.24** | 29.89 | 5.99 | 18.19 | 31.37 | **8.50** | 20.65 | **21.20** |
| *Truncating input tokens to the pretraining context length* | | | | | | | | | | | |
| Llama2-7b-chat | 4k | 37.15 | 9.47 | 18.05 | 29.75 | 6.61 | 18.96 | 28.75 | 6.24 | 19.37 | 19.37 |
| Llama2-13b-chat | 4k | 37.27 | 9.79 | 18.49 | **30.49** | **6.69** | **19.23** | **29.63** | **6.54** | **19.65** | **19.75** |
| Vicuna1.3-7b | 2k | 34.63 | 8.73 | 16.87 | 29.01 | 6.28 | 18.18 | 24.18 | 4.93 | 15.93 | 17.63 |
| Longchat-7b-16k | 2k | 37.01 | 9.61 | 18.21 | 26.45 | 5.05 | 16.88 | 23.92 | 4.65 | 15.75 | 17.50 |
| Chatglm2-6b-8k | 2k | 36.91 | 9.45 | 17.96 | 27.74 | 5.77 | 17.62 | 25.92 | 5.61 | 17.57 | 18.28 |
| XGen-7b (2k-4k-8k) | 2k | **37.72** | **9.97** | **18.77** | 28.21 | 5.94 | 18.69 | 26.94 | 5.92 | 18.24 | 18.93 |
| *Truncating input tokens to the further finetuning context length* | | | | | | | | | | | |
| Chatglm-6b-32k | 32k | 32.65 | 8.09 | 16.51 | 22.05 | 6.10 | 16.61 | **28.94** | **8.86** | **20.83** | 17.84 |
| Longchat1.5-7b-32k | 32k | 32.49 | 7.79 | 15.97 | 27.53 | 5.80 | 17.94 | 25.29 | 5.22 | 16.49 | 17.16 |
| Longchat-7b-16k | 16k | 35.05 | 8.57 | 16.70 | 26.07 | 5.97 | 17.06 | 20.13 | 4.74 | 13.21 | 16.38 |
| Vicuna1.5-7b-16k | 16k | 36.84 | 9.78 | 17.66 | 28.91 | 6.47 | 18.25 | 26.90 | 5.53 | 17.33 | 18.63 |
| Longchat-13b-16k | 16k | 34.41 | 8.07 | 16.45 | 27.24 | 5.63 | 17.00 | 24.58 | 5.85 | 16.32 | 17.28 |
| Vicuna1.5-13b-16k | 16k | 36.30 | 8.69 | 18.20 | 28.59 | 6.15 | 18.49 | 27.82 | 6.39 | 18.83 | 18.82 |
| Llama2-13b-NTK* | 16k | 35.22 | 8.53 | 17.04 | 23.97 | 4.72 | 14.89 | 18.92 | 4.13 | 13.16 | 15.61 |
| Llama2-13b-NTK(Dyn)* | 16k | 28.89 | 7.21 | 14.83 | 26.86 | 5.33 | 17.55 | 22.29 | 4.88 | 15.29 | 15.90 |
| Longchat-13b-16k | 8k | 34.29 | 8.21 | 16.06 | 26.76 | 5.61 | 16.77 | 20.86 | 4.01 | 13.81 | 16.26 |
| Chatglm2-6b-8k | 8k | **38.07** | **9.61** | 17.33 | **29.81** | **6.47** | 18.91 | 24.74 | 4.45 | 4.44 | 18.36 |
| XGen-7b-8k | 8k | 35.94 | 8.49 | **17.92** | 28.92 | 6.28 | **19.11** | 28.06 | 6.12 | 19.17 | **18.89** |
| MPT-7b-65k | 8k | 15.91 | 2.91 | 11.18 | 7.66 | 1.00 | 7.00 | 5.24 | 0.71 | 5.10 | 6.30 |

Table 7: Performance of various models on **query-based** summarization and generation tasks in terms of ROUGE.

QA tasks. As for query-based tasks that pose questions from a specific perspective, performance can be significantly degraded if the instruction isn't fully understood. As we mentioned before, the increased input length can also lower the model's ability to comprehend lengthy instructions, thereby inhibiting its capability to generate answers that closely match the length of the ground truth. This phenomenon is less likely to be observed with more sophisticated LLMs (i.e. Turbo-16k). A naive solution is adding the instruction at both the beginning and end of the long input but there is still room to improve the ability of instruction understanding for LCLMs.

**Retrieve-based models vs long context models**  We compare a representative LCLM baseline Turbo-16k-0613 with its short version Turbo-4k-0613 but enhanced with retrieval in Table 3(closed-ended tasks) and Table 4(open-ended tasks). We use a sparse retrieval retriever bm25 and a strong dense retriever text-embedding-ada-002. Retrieve-based approaches generally yield better outcomes for tasks that have readily retrievable answers. For example, for long lectures understanding where the long document always contains some definitions and explanations for some academic terms, retrieval-based approaches obtain better results. However, retrieval is not a general solution as its performance is strongly related to instruction and document style. For example, they would never answer questions like *how many sentences are there in a document.* Our results show that **CodeU** and **GSM(16-shot)** in L-Eval can not be solved by retrieval. Retrieval-based methods also face difficulties in automatically **identifying the query** from user inputs. Retrieval methods demonstrate comparatively less satisfactory performance in tasks where the answer cannot be retrieved, such as topic retrieval or tasks that demand models with long-range reasoning abilities like financial QA. Retrieve-based models produce similar or even superior results for summarization tasks. This may be because some paragraphs resembling summaries can be retrieved. Besides, we also noticed that the main reason why regular Turbo-0613 outperforms Turbo-16k is its superior ability to accurately follow instructions. However, even for these tasks, there are instances where the predicted answer might be "I don't know" or "not mentioned" due to the limitation of the retrieval process. When

| Model | Tokens | gov_report | | | news | | | patent | | | tv_show | | | Avg |
|---|---|---|---|---|---|---|---|---|---|---|---|---|---|---|
| | | R-1 | R-2 | R-L | R-1 | R-2 | R-L | R-1 | R-2 | R-L | R-1 | R-2 | R-L | |
| Turbo-16k-0613 | 16k | **45.9** | **15.6** | **23.6** | 35.3 | 8.1 | 16.1 | **46.0** | **20.3** | **29.3** | 32.0 | 5.4 | 16.9 | **24.5** |
| AdaEmb-Turbo-0613 | 4k | 45.0 | 14.3 | 20.8 | 35.7 | 7.7 | 15.4 | 45.6 | 15.9 | 27.6 | 30.0 | 3.3 | 15.2 | 23.0 |
| bm25-Turbo-0613 | 4k | 44.6 | 14.2 | 21.5 | **38.4** | **9.1** | **16.8** | 43.3 | 15.5 | 27.1 | 31.0 | 4.6 | 15.4 | 23.4 |
| *Truncating input tokens to the pretraining context length* | | | | | | | | | | | | | | |
| llama2-7b-chat | 4k | 43.7 | 15.3 | 22.2 | 33.2 | 6.4 | 15.5 | 49.2 | 22.9 | 31.6 | 29.4 | 4.8 | 15.6 | 24.1 |
| llama2-13b-chat | 4k | **46.3** | 16.1 | 24.0 | **34.9** | **8.1** | **16.3** | 48.4 | 20.9 | 30.3 | **32.6** | **6.6** | **17.2** | **25.1** |
| vicuna1.3-7b | 2k | 44.6 | 16.4 | 23.2 | 32.9 | 6.9 | 14.8 | 44.7 | 20.4 | 28.8 | 28.7 | 3.6 | 14.8 | 23.3 |
| longchat-7b-16k | 2k | 43.6 | 16.2 | 23.7 | 28.1 | 4.8 | 13.0 | 47.0 | 22.2 | 30.9 | 27.2 | 3.0 | 14.4 | 22.8 |
| chatglm2-6b-8k | 2k | 45.2 | **18.3** | **24.6** | 32.1 | 6.9 | 15.0 | 44.6 | 22.1 | 30.0 | 26.4 | 2.6 | 13.8 | 23.4 |
| xgen-7b-8k | 2k | 45.1 | 17.2 | 22.9 | 35.0 | 7.5 | 15.5 | **49.6** | **25.2** | **34.6** | 28.8 | 3.6 | 15.4 | 23.9 |
| *Truncating input tokens to the further finetuning context length* | | | | | | | | | | | | | | |
| chatglm2-6b-32k | 32k | 38.1 | 16.1 | 21.0 | 24.2 | 5.8 | 12.8 | 46.5 | 24.1 | **32.5** | 23.4 | 4.2 | 13.8 | 21.8 |
| longchat1.5-7b-32k | 32k | 45.7 | 17.7 | 24.0 | **36.8** | **8.7** | 15.7 | 42.0 | 18.2 | 27.2 | 21.5 | 2.7 | 13.0 | 22.7 |
| longchat-7b-16k | 16k | 47.2 | 18.9 | 23.9 | 27.7 | 5.4 | 13.4 | 46.2 | 20.9 | 30.1 | 26.2 | 3.3 | 14.7 | 23.1 |
| vicuna1.5-7b-16k | 16k | 47.2 | 18.9 | 25.0 | 32.3 | 6.8 | 15.5 | **48.1** | **25.1** | 32.4 | 26.0 | 3.6 | 14.8 | **24.6** |
| longchat-13b-16k | 16k | 46.2 | 18.2 | 24.1 | 35.2 | 7.6 | 15.8 | 45.3 | 22.6 | 29.8 | **31.9** | **6.0** | **17.3** | 24.0 |
| vicuna1.5-13b-16k | 16k | 45.2 | 17.9 | 24.2 | 31.6 | 6.8 | 15.2 | 46.1 | 21.8 | 30.0 | 28.3 | 3.7 | 16.3 | 23.9 |
| llama2-13b-NTK | 16k | 33.0 | 11.0 | 17.7 | 26.0 | 6.4 | 13.5 | 37.9 | 13.5 | 22.9 | 25.6 | 5.3 | 14.0 | 18.9 |
| llama2-13b-NTK(Dyn) | 16k | 42.0 | 14.9 | 22.4 | 34.0 | 7.8 | **15.9** | 45.3 | 19.1 | 28.5 | 25.5 | 3.9 | 13.9 | 22.7 |
| longchat-13b-16k | 8k | **49.3** | **19.5** | **25.1** | 34.9 | 7.4 | 15.5 | 43.5 | 20.1 | 28.0 | 31.0 | 4.5 | 15.7 | 24.5 |
| chatglm2-6b | 8k | 40.6 | 14.3 | 21.5 | 32.9 | 7.2 | 15.1 | 46.3 | 22.3 | 31.4 | 27.5 | 2.6 | 14.5 | 23.0 |
| xgen-7b-8k | 8k | 40.2 | 13.8 | 21.1 | 31.9 | 6.0 | 15.3 | 45.9 | 21.4 | 29.2 | 28.2 | 3.3 | 15.2 | 22.6 |
| mpt-7b-65k | 8k | 33.3 | 10.7 | 19.3 | 13.6 | 1.5 | 9.2 | 25.5 | 12.2 | 20.2 | 11.0 | 1.3 | 6.4 | 13.6 |

Table 8: Performance of various models on long document summarization tasks in terms of ROUGE.

evaluating retrievers, bm25 often matches the performance of the dense retriever, ada-embedding, in closed-ended tasks. However, in the open-ended tasks, the dense retriever ada-embedding outperforms BM25 by more than two points. This superior performance can be attributed to the dense retriever's ability to leverage not only term matching but also semantic matching.

**Dynamic NTK scaling rules do not hold in practical tasks** Dynamic NTK-aware positional embedding LocalLLaMA (2023a) is becoming more and more popular for extrapolation without further training. Based on dynamic NTK, given an input sequence with length $L$ and the model pertaining length $l$, we can set the original base 10,000 in RoPE to $10,000 \times \frac{L}{l}^{\frac{d}{d-2}}$ where $d$ is the head dimension, if we want to adapt the model to the longer context length $L$. We find that the scaling rule does not hold in practical tasks when the number of input tokens changes. The improvements can be further improved if using some variants of NTK. We study 2 simple modifications on the original dynamic NTK: (1) NTK-bias which

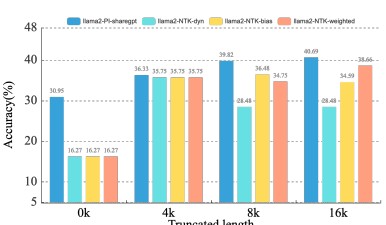

Figure 6: Peformance of different NTK-based methods when tackling input length at multiple scales.

means we use the base $10,000 \times (\frac{L}{l} + 1)^{\frac{d}{d-2}}$ where 1 is the bias (2) NTK-weighted which means we use the base $10,000 \times (\frac{L}{l} * 2)^{\frac{d}{d-2}}$. Results are shown in Figure 6 where Llama2-PI-sharegpt is a fine-tuned baseline using position interpolation. We test the results of 4 models by truncating the input length of test cases in Coursera. We can observe that employing which variants of NTK are strongly affected by the maximum tokens of the dataset. When the input length is between 4k and 8k, NTK+bias gets the best results and NTK-weighted baseline is more robust on 16k input tokens.

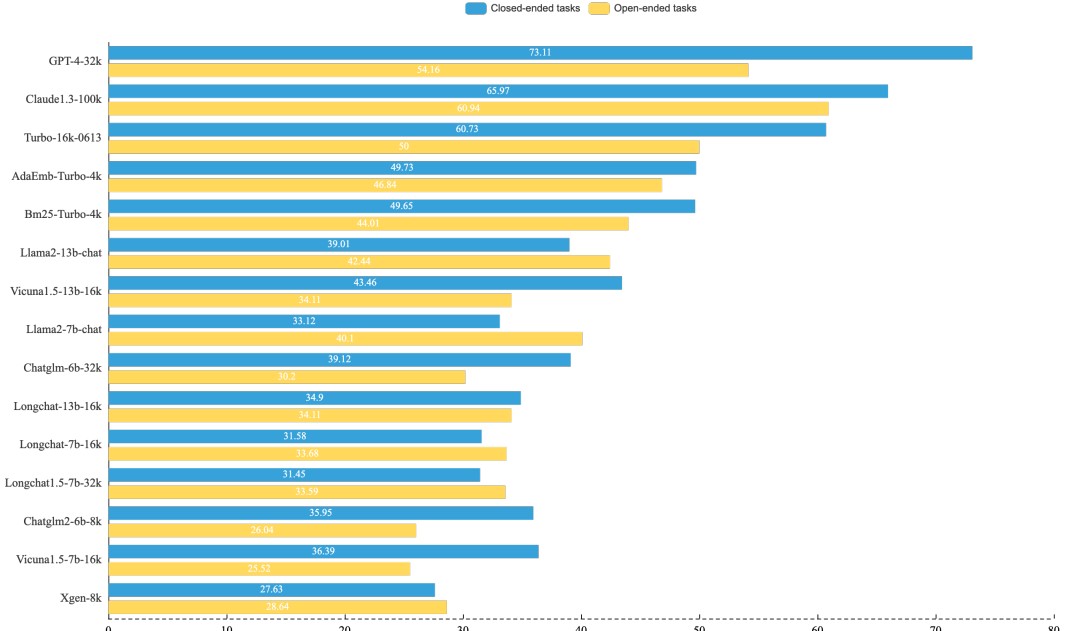

Figure 7: Overall results on open-ended tasks and closed-ended tasks. We find that GPT-4-32k is more capable of closed-ended tasks demonstrating powerful reasoning ability over long context since most closed-ended task in L-Eval has less than 32k input tokens, but the 100k context length help Cluade surpass both GPT-4-32k and Turbo-16k on open-ended tasks which generally has more input tokens.

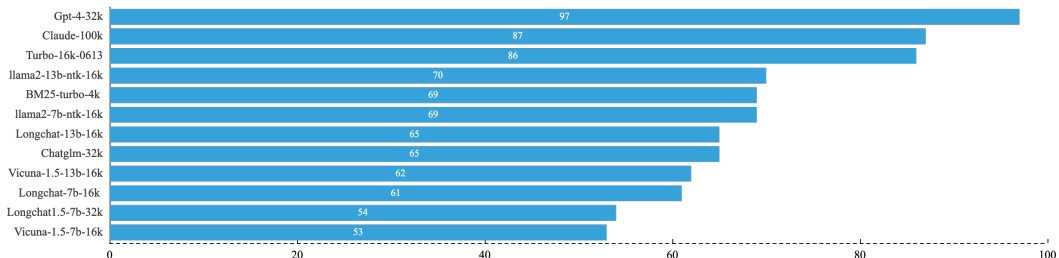

Figure 8: Overall results on the topic retrieval tasks. Testing short context models on this task with truncated input texts is unfair, so we only include long context LLMs.

## B  DATA COLLECTION AND ANNOTATION FOR L-EVAL

In our pursuit of diverse, comprehensive, and relevant data, we sourced datasets from a wide array of platforms and sources. These datasets, represent various facets of everyday life and specialized fields and present different challenges for LCLMs. We leveraged resources from previous open-source datasets, Coursera subtitles, earning call transcripts from corporate websites, GitHub, etc. The instruction styles in L-Eval include multiple-choice questions, school math with many examples, key topics retrieval from lengthy dialogues, text summarization, and abstractive question answering, encompassing a wide range of tasks. The construction of each dataset and our effort to make it more challenging are as follows.

### B.1  TOFEL (ENGLISH TESTS)

This dataset is sourced from the TOEFL Practice Online and we collect the data from TOEFL-QA (Tseng et al., 2016; Chung et al., 2018) and all lectures from a single TPO have been consolidated into one lengthy lecture. After the consolidation, we select the top 15 longest lectures.

```
Example 1

Input:  <Multiple long lectures> \n\n
Question:  why did Frantzen go to the sales barn
A. to study human form and movement
B. to earn money by painting portraits
C. to paint farm animals in an outdoor setting
D. to meet people who could model for her painting
\n\n Answer:
Ground truth:  A
```

### B.2  GSM(16-SHOT)(GRADE SCHOOL MATH)

This dataset is derived from 100-grade school math problems in the GSM8k dataset (Cobbe et al., 2021). Increasing the number of high-quality and complex examples usually has a positive effect on solving math problems. We construct 16 in-context examples with lengthy Chain-of-thought for this task where 8 examples come from *chain-of-thought-hub*[16] using the hardest prompt and the remaining 8 examples are constructed by us. We selected 8 questions from GSM8k based on their difficulty and annotated the solving process. Models with 2k or 4k context length face difficulties while encoding the 16 examples. We experiment with the newly constructed examples and it perform better than only encoding 8 examples. Concretely, the accuracy rises from 79 (8-shot) to 84 (16-shot) when using turbo-16k-0613 as the base model.

```
Example 2

Input:  <example 1> \n\n <example 2> \n\n ...  <example n> \n\n
Question:  Janet's ducks lay 16 eggs per day.  She eats three for breakfast every
morning and bakes muffins for her friends every day with four.  She sells the remainder
at the farmers' market daily for $2 per fresh duck egg.  How much in dollars does she
make every day at the farmers' market?  \n\n
Let's think step by step
Ground truth:  18
```

### B.3  QUALITY (GUTENBERG)

This dataset is sourced from the multiple choice QA dataset QuALITY (Pang et al., 2022) which contains multiple-choice questions derived from the literature on Gutenberg. We filter 20 long stories and 202 questions and correct/delete questions with annotation errors. We found that most questions in QuALITY can be solved by extracting paragraphs from long texts. We further enhance some

---

[16]https://github.com/FranxYao/chain-of-thought-hub/blob/main/gsm8k/lib_prompt/prompt_hardest.txt

synthesis questions that need a global understanding of the document. Examples of the annotated synthesis questions are as follows:

1. *What can we infer from the longest sentence in the story?*
2. *The longest dialogue is spoken by whom?*
3. *Extract names mentioned in the longest sentence in the story.*
4. *How many words are there in the story?*
5. *How many sentences are there in the story?*

The reference source sentences are automatically located and the ground truth answers are manually annotated by us. An example of the original question in QuALITY is like this:

---

**Example 3**

```
Input:  <A long story>\n\n
Instruction:  Why did Syme accept the mission with Tate?
(A) He needed a way back to Earth
(B) He felt he would collect a reward along the way
(C) He respected Tate
(D) He had no plan for his life, so he jumped on the adventure
Ground truth:  (B) He felt he would collect a reward along the way
```

---

### B.4 COURSERA (ADVANCED LECTURES)

This dataset originates from the Coursera website[17]. We selected and completed 4 courses:

1. *Ask Questions to Make Data-Driven Decisions*,
2. *Data Scientist's Toolbox*,
3. *Process data from dirty to clean*,
4. *Improving Deep Neural Networks: Hyperparameter Tuning, Regularization and Optimization*.

The input long document is the subtitles of the videos and we merge courses in one week into one single long lecture. Questions and the ground truth answers are labeled by the authors. The instruction style of Coursera takes the format of multiple choice. In order to increase the difficulty of the task, we have set **multiple correct options**. Failure to select all correct choices will result in receiving only a quarter of the total points for that question.

---

**Example 4**

```
Input:  <A long lecture>\n\n
Question:  When working with a new team, which of the following actions can help you to
adapt to different communication expectations?  Select all that apply.
A. Ask questions when you are unsure of something
B. Learn the team's preferred communication style
C. Observe how teammates communicate with each other
D. Ignore the team's communication preferences and use your own style
\n\n Answer:
Ground truth:  ABC
```

---

### B.5 SFCITION (SCIENTIFIC FICTIONS)

We annotate this sub-task to test the loyalty of the LCLM to the input context. LLMs have acquired many commonsense in their pertaining corpus known as parametric knowledge (Wang et al., 2023). However, we argue that in LCLMs, contextual knowledge is more crucial than parametric knowledge. In real-world applications, many long documents are private and can never be seen during pretraining. It may contain new knowledge or describe a new world which may be opposite

---

[17]https://coursera.org/

to the pretraining knowledge. The language model should follow contextual knowledge instead of parametric knowledge. To simulate this scenario, we annotate a science fiction dataset consisting of True or False questions. The original works are sourced from SFGram[18]. We manually select documents that fit our experimental conditions and annotate them with questions and corresponding answers. Most of the answers to these questions contradict real-world principles and do not comply with actual physical laws, such as the statement: *Humans have invented the time machine.* As a result, open-source models have very serious hallucination problems which in turn help them acquire a high score on this dataset. So we also give the answer based on real-world knowledge, and the final accuracy is calculated by the average of loyalty and factuality.

```
Example 5

Input:  <A scientific fiction>\n\n
Question:  We cannot get to the centre of the Earth, True or False?  Answer this
question based on the world described in the document.
Ground truth:  False

Question:  We cannot get to the centre of the Earth, True or False?  Answer this
question based on the real-world knowledge and facts up until your last training.
Ground truth:  True
```

### B.6  CODEU (PYTHON)

This dataset is used to test the capability of understanding long code. Given a lengthy code base, we will call some functions defined in the codebase and the model should infer the final output of the program. We mainly use source code from Numpy[19]. We also write a string processing codebase containing more than 100 functions that take a string as input such as extracting the email address from an input string. To prevent LLMs from answering the question based on their parametric knowledge, We replace the original function name defined in Numpy with `Op1, Op2..., OpN`. The Language Model (LLM) should first identify where the function is called and determine which functions are invoked, ultimately ascertaining the results of the operations. CodeU represents the most challenging task within L-Eval. Even the most potent model, GPT-4-32k, achieves an accuracy of only 25.55%.

```
Example 6

Input:  <The beginning of a lengthy Python program>
def Op1():  ...
def Op2():  ...
args = [4,5,6]
output = Op1(args)
print(output)
<The rest of the program>\n\n
Instruction:  What is the output of this program?  Please carefully read through these
code snippets and comments.  You should first identify where the functions are defined
and then figure out what they do.
\n\n let's think step by step:
Ground truth:  [1,2,3]
```

### B.7  TOPICRET (LENGTHY CONVERSATION)

This dataset comes from the LongChat repository (Li et al., 2023a)[20], and its task style focuses on retrieving topics from extensive chat histories. Recent studies show that language models are good at retrieving information from the very beginning or end of its input context but are usually lost in the middle (Liu et al., 2023). To make the task more challenging, we enhance the original task by asking the model to extract **the second and the third** topic.

---

[18]https://github.com/nschaetti/SFGram-dataset
[19]https://github.com/numpy/numpy
[20]https://github.com/DachengLi1/LongChat

---

**Example 7**

```
Input:  <A long conversation > \n\n
Question:  What is the second topic we discussed?  Only give me the topic name.  Do not
summarize yourself.
Ground truth:  The future of space tourism
```

---

## B.8   LONGFQA (FINANCE)

We find that there is a lack of long open-ended QA datasets in finance. The long context finance dataset is derived from earnings call transcripts obtained from the *Investor Relations* section of the company websites. We annotate 6 transcripts from 6 different incorporations, Lumentum Oclaro[21], Theragenics[22], FS KKR Capital Corp[23], LaSalle Incorporated[24], Renewable Energy Group[25] with 54 questions based on these transcripts.

---

**Example 8**

```
Input:  <A long document>\n\n
Instruction:  You are asked to act as a member of the Financial Results Conference Call
and answer the question:  What major actions has Greg Dougherty, the CEO of Oclaro,
highlighted as being undertaken by the company for its restructuring plan?  \n Answer
this question with xx words.
Ground truth:  Oclaro has been implementing a significant restructuring plan, which
includes closing our second major...
```

---

## B.9   CUAD (LAW)

Questions on the Legal domain are drawn from the CUAD (Contract Understanding Atticus Dataset) dataset (Hendrycks et al., 2021b) designed for supporting NLP research for automating legal contract review. We manually filter 20 documents with annotated QA pairs from CUAD.

---

**Example 9**

```
Input:  <Legal contracts> \n\n
Instruction:  Highlight the parts (if any) of this contract related to Ëxpiration
DateËhat should be reviewed by a lawyer.  Details:  On what date will the contract's
initial term expire?  \n Answer this question with xx words.
Ground truth:  The term of this Agreement shall commence on the Effective Date and
shall continue in full force and effect for an initial period of five (5) years.
```

---

## B.10   MULTIDOC2DIAL (DIALOGUES OVER MULTI-DOCUMENTS)

This dataset is sampled from the MultiDoc2Dial dataset (Feng et al., 2021) which aims to model goal-oriented dialogues grounded in multiple documents. It contains dialogues from 4 different domains: Finance, Travel, Entertainment, and Shopping. Each dialogue in the dataset is grounded in 2-5 relevant documents covering different topics within the domain.

---

**Example 10**

```
Input:  <Multiple long documents> \n\n
Instruction:  How long will Driver's Ed courses be valid for?  \n Answer this question
with xx words.
Ground truth:  For roughly 1 one year.  Maybe longer depending on the course.
```

---

[21] https://investor.lumentum.com/overview/default.aspx
[22] https://www.sec.gov/Archives/edgar/data/
[23] https://www.fskkradvisor.com/investor-relations/
[24] https://ir.jll.com/overview/default.aspx
[25] https://www.regi.com/resources/press-releases

## B.11 NATURAL QUESTIONS (WIKIPEDIA)

We filter 20 wikipedia long documents from Natural Question (Kwiatkowski et al., 2019) on Google Research datasets. Questions can be answered with the same documents are merged, and duplicate questions are removed.

> **Example 11**
>
> **Input:** <Documents from Wiki>\n\n
> Instruction: when did season 2 of handmaid's tale start? \n Answer this question with xx words.
> **Ground truth:** April 25, 2018

## B.12 NARRATIVEQA (NARRATIVES)

This dataset is collected from NarrativeQA (Kočiský et al., 2017) which has the longest document length in L-Eval. The original question-answering dataset was created using entire books from Project Gutenberg[26] and movie scripts from various websites. Summaries of the books and scripts were taken from Wikipedia and given to annotators. Our work focuses on correcting the annotation error for example, there are some issues where the main character in the question does not even appear in the input document at all.

> **Example 12**
>
> **Input:** <A long novel>\n\n
> Instruction: Why did Mary pay off the debt for Ann's family? \n Answer this question with xx words.
> **Ground truth:** Mary was in love with Ann.

## B.13 QASPER (PAPERS)

This dataset is filtered from the Qasper dataset (Dasigi et al., 2021), which is a question-answering resource focused on NLP papers. The dataset was constructed using NLP papers that were extracted from the Semantic Scholar Open Research Corpus (S2ORC). After filtering, we remove the unanswerable questions and the extractive version answers. We also discovered instances where identical questions yielded contradictory answers. We addressed this issue by meticulously reviewing the paper and rectifying the incorrect responses.

> **Example 13**
>
> **Input:** <A long paper>\n\n
> Instruction: How did they obtain the dataset? \n Answer this question with xx words.
> **Ground truth:** public resources where suspicious Twitter accounts were annotated, list with another 32 Twitter accounts from BIBREF19 that are considered trustworthy.

## B.14 OPENREVIEW (PAPERS)

This task aims to help researchers working on scientific papers by dealing with tasks like correcting grammar errors or typos and writing some sections. We include 3 tasks in the paper writing assistant task of L-Eval: 1) writing an Abstract section, (2) writing a Related Work section, and (3) finally giving a review of this paper including valuable suggestions and questions. Notably, we discourage reviewers from using large models for reviews. Our aim is to assist authors in further improving their papers. Therefore, we ask the model to give some valuable suggestions and raise some questions for authors. We filter 20 papers with well-written reviews for L-Eval. We use the processed PDF files from Yuan et al. (2021).

---

[26]https://www.gutenberg.org

---

**Example 14**

**Input:** `<A long paper>\n\n`
`1. Instruction: Please generate the Abstract section for this paper. \n Answer this`
`question with xx words.`
`2. Instruction: Please summarize related work and you should include the following`
`works [a list of papers]. \n Answer this question with xx words.`
`3. Instruction: Please write a review for this paper and you should provide some`
`suggestions and raise some questions in your review. \n Answer this question with xx`
`words.`
**Ground truth:** `Conventional out-of-distribution (OOD) detection schemes based on`
`variational autoencoder or Random Network Distillation (RND) have been observed to`
`assign ...`

---

## B.15  GovReport (Government Reports)

This dataset is filtered from the government report summarization dataset (Huang et al., 2021), the dataset consists of long reports written by U.S. government research agencies such as the Congressional Research Service and Government Accountability Office. The documents and summaries in this dataset are longer compared to other long document summarization datasets. We manually filter 13 documents with human-written summaries from the original dataset.

---

**Example 15**

**Input:** `<A government report>\n\n`
`Instruction: Please help me summarize this government report. \n Answer this question`
`with xx words.`
**Ground truth:** `The President of the United States has available certain powers that`
`may be exercised in the event that the nation is threatened by crisis, exigency, or`
`emergency circumstances...`

---

## B.16  QMSum (Meetings)

This dataset sourced from the QMSum (Zhong et al., 2021), this dataset contains query-based meeting summarizations. Query-based summarization aims to summarize the document given a specific aspect. We selected 20 meeting transcripts accompanied by queries, specifically choosing those that could not be easily addressed through retrieval methods.

---

**Example 16**

**Input:** `<Meeting trancripts>\n\n`
`Instruction: What was agreed upon on sample transcripts? \n Answer this question with`
`xx words.`
**Ground truth:** `To save time, speaker mn005 will only mark the sample of transcribed`
`data for regions of overlapping speech, as opposed to marking all acoustic events...`

---

## B.17  SPACE (Reviews)

The review (opinion) summarization aims to summarize the reviews from customs reviews on a restaurant or hotel. We obtain 20 samples from the validation and test set of SPACE (Angelidis et al., 2021) where human-written abstractive summaries are created for 50 hotels based on 100 input reviews each. SPACE consists of customer reviews of hotels from TripAdvisor, with 1.1 million training reviews for 11,000 hotels. The original task asks the model to summarize hotels from multiple aspects: food, location, cleanliness, etc. We construct the instructions for review summarization with GPT-4 and some examples.

> **Example 17**
>
> **Input:** <Multiple reviews>\n\n
> Instruction: Give a broad summary of guest impressions about Doubletree by Hilton
> Seattle Airport. \n Answer this question with xx words.
> **Ground truth:** The staff are friendly and exceptional. Every room (lobby included) was
> very clean. They are spacious, very quiet, and come with a coffee maker...

### B.18 MULTI-NEWS (NEWS)

This dataset is sourced from the Multi-News (Fabbri et al., 2019). The original Multi-News dataset
contains news articles as well as human-written summaries of those articles, compiled from the
website newser.com where each article consists of multiple short news articles. We select 10 articles
for the L-Eval benchmark.

> **Example 18**
>
> **Input:** <News articles>\n\n
> Instruction: Please summarize these news articles. \n Answer this question with xx
> words.
> **Ground turth:** Why did Microsoft buy Nokia's phone business? We now know Microsoft's
> answer: The computing giant released a 30-slide presentation today arguing that the
> move will improve Microsoft...

### B.19 BIGPATENT (PATENTS)

This dataset is derived from the BigPatent (Sharma et al., 2019) project, which consists of 1.3 million
records of U.S. patent documents along with human-written abstractive summaries, we select 13
patent patents from the original dataset.

> **Example 19**
>
> **Input:** <A long patent>\n\n
> Instruction: You are a patent examiner. Please write a summary of this patent. \n
> Answer this question with xx words.
> **Ground truth:** The invention provides a method and system for cleaning pet paws by
> providing a bounded container containing...

### B.20 SUMMSCREEN (TV SHOW)

This dataset originates from the SummScreen (Chen et al., 2022), the original dataset is an abstrac-
tive summarization dataset combining TV series transcripts and episode recaps. SummScreen is
constructed from fan-contributed websites. We use 13 of these transcripts in L-Eval.

> **Example 20**
>
> **Input:** <TV series transcripts> \n\n
> Instruction: Write a summary of the scene. \n Answer this question with xx words.
> **Ground turth:** Feeling guilty over Phoebe missing out on London, the gang plans a
> weekend trip to Atlantic City, but just as they are about to leave...

