# OpenReview forum: "L-Eval: Instituting Standardized Evaluation for Long Context Language Models"
_ICLR.cc/2024/Conference — Submitted to ICLR 2024_

### Official Review · Reviewer_4r4K · 2023-10-31

**Soundness:** 3 good
**Presentation:** 3 good
**Contribution:** 3 good
**Rating:** 6
**Confidence:** 3

**Summary:**

The authors introduce a carefully curated long context evaluation benchmark with both open and closed tasks. They also propose an LLM-judge evaluation method that correlates better with human judgment than traditional N-gram metrics that other benchmarks still use. Their analysis provides insights into the types of prompt settings, tasks, and evaluations that work well for this problem.

**Strengths:**

* Carefully curated and manually chosen questions that are unintuitive to models that do not read and understand the context (e.g. scifi that contradicts real world physics)
* Targets domains that are understudied, e.g. long context finance questions
* LIE shows interesting analysis → what details matter for prompting and evaluating long contexts like this.
* Continued finetuning analysis between open and closed tasks are insightful
* The effects of NTK positional embeddings on retrieval vs reasoning (their negative correlation) is very interesting

**Weaknesses:**

* Uses Claude-100k as the data filter. This detects unanswerable questions, but careful human review would be better, rather than steering scores towards closed LLMs.
* Length Instruction Enhanced (LIE) is arguably not a new or significant contribution, so the authors should be careful not to frame it as such. The analysis of it is what is interesting.

**Questions:**

* Did you test performance of models with the question but without the context? I.e. you make the argument that the questions are designed not to be easy just from using parametric knowledge – can we validate this?
* Confused about the 96 subset. Are there many evaluation questions in your benchmark ignored / not used?

---

> ### Author Response · Authors · 2023-11-14
> **Responses to reviewer 4r4K**
>
> We appreciate the time and effort you've invested in reviewing this paper. We summarize your primary concerns to be as follows:
>
> > **Q1**: Human review for the dataset to avoid steering scores towards closed LLMs.
>
> Thank you for your valuable advice. We use Claude-100k to detect unanswerable questions because it has the longest context window and is powerful in our preliminary experiments. We plan to re-detect the unanswerable questions using another long context model, gpt3.5-turbo-16k, and review the questions to further ensure the quality.
>
> > **Q2**: Issues regarding length-instruction-enhanced evaluation.
>
> We concur that the enhancement of length instruction is not a significant contribution of this work. Rather, our primary objective is to draw researchers' attention to the length bias in evaluation metrics. The evaluation metrics for LCLMs should favor accurate content over precise length. We will further clarify this point in the introduction section of our paper.
>
> > **Q3**: Test the model without the context.
>
> Thank you for a very interesting question. We test 3 models on the 4 new datasets annotated by us: SFiction (factuality+loyalty) and Coursera (accuracy), CodeU (accuracy), and LongFQA (F-1). Results are as follows:
>
> |                   | SFiction | SFiction(no ctx) | Coursera | Coursera(no ctx) |
> |-------------------|---------:|-----------------:|---------:|-----------------:|
> | GPT3.5-turbo-16k  |    64.84 |            50.78 |    63.51 |            44.47 |
> | Llama2-13b-4k     |    54.68 |            42.18 |    35.75 |            16.27 |
> | Vicuna1.5-13b-16k |    61.71 |            46.09 |    40.69 |            30.75 |
>
> |                   | CodeU | CodeU(no ctx) | LongFQA | LongFQA(no ctx) |
> |-------------------|------:|--------------:|--------:|---------------:|
> | GPT3.5-turbo-16k  | 12.22 |           0.00|    45.36|           34.89|
> | Llama2-13b-4k     |  1.11 |           0.00|    38.07|           20.42 |
> | Vicuna1.5-13b-16k |  0.00 |           0.00|    45.57|           18.90 |
>
> We will include more models in our camera-ready version.
>
> > **Q4**: Issues regarding the 96-question subset
>
> We selected the 96-question subset, which is similar in scale to the popular Vicuna 80-question test set.  We split the subset mainly due to the high cost of GPT-4 evaluation. We have also included the results of SPACE, QMSum, NQ, and NarrativeQA with GPT-4 evaluation in Table 2, but testing the 4 datasets has already exceeded a cost of **$100**. Therefore, we only reported n-gram metrics results in Tables 6, 7, and 8 for the remaining open-ended tasks. Researchers can conduct experiments on some sub-tasks from L-Eval or directly on the 96-question subset based on their settings.

---

> ### Author Response · Authors · 2023-11-21
> **Looking forward to receiving your feedback**
>
> Dear Reviewer 4r4K,
>
> Thank you for your valuable time to review our work and constructive feedback. We sincerely thank you for thinking the findings in this paper interesting and we hope that our revisions and clarifications address your concerns satisfactorily.
> If there are any additional questions, we will respond to them as soon as possible.
>
> Best wishes,
>
> The Authors

---

### Official Review · Reviewer_cNRy · 2023-11-02

**Soundness:** 2 fair
**Presentation:** 3 good
**Contribution:** 2 fair
**Rating:** 5
**Confidence:** 3

**Summary:**

This paper presents L-Eval, a dataset for evaluating long context language models (LCLMs). The authors constructed an evaluation suite featuring 20 sub-tasks, 508 long documents, and over 2,000 human-labeled query-response pairs showcasing varied question styles and domains. Apart from this, the paper raises concern over the efficacy of prevalent ngram matching metrics as they often fail to sync with human judgment. Therefore, it favors employing length-instruction-enhanced (LIE) evaluation and using large language model (LLM) judges. A study of 4 widely-used commercial LLMs and 12 open-source models, evaluated with the introduced L-Eval benchmark, provides findings for future development of LCLMs evaluation methodologies.

**Strengths:**

1. The author clearly articulates the intended content, including methods and procedures for dataset construction, statistical features of the dataset, evaluation methods, and so on.
2. The problem, namely the "Evaluation for Long Context Language Models" is important. The proposed dataset and conclusions regarding existing evaluation methods contribute to the advancement of long context LLMs.
3. A comprehensive evaluation has been carried out by the author on current commercial and open-source LLMs.

**Weaknesses:**

1. The key contribution of this paper is the dataset it provided, but the authors do not offer sufficient analysis on the data collection and construction methods. For instance, in section 3.1 detailing the construction of the Coursera dataset, the authors mention, "In order to increase the task's difficulty, we have set multiple correct options. To the best of our knowledge, this is the first multi-choice dataset with multiple correct answers, and it is more challenging than single-option questions." Could it be because without this increased difficulty, there wouldn't be any differentiation among various models?
2. The findings of the paper are rather intuitive and not enough insightful. For example, the authors mention in the penultimate paragraph of the Introduction, "There is still a significant gap between open-source LCLMs and commercial models, for both closed-ended tasks (Table 3) and open-ended tasks evaluated by LLMs and humans (Table 4, 5). However, this gap is not accurately reflected by n-gram metrics."

**Questions:**

1. Could you elaborate more precisely on the motivations and insights regarding the dataset construction methods, such as Coursera, CodeU, and LongFQA?
2. Could the authors provide further insights into the shortcomings of current open-sourced long-context models, and how the dataset provided in this paper could facilitate enhancements in these problems?

---

> ### Author Response · Authors · 2023-11-14
> **Responses to reviewer cNRy**
>
> We appreciate your comments and review of the paper. We summarize your key concerns as follows:
>
> > **Q1**: Motivations regarding to dataset construction, especially the datasets annotated from scratch.
>
> *General motivations for dataset annotation and correction:*
>
> (1) Previous large-scale long sequence datasets are of varying annotation quality. Considering the overhead of LCLMs decoding, we need diverse and high-quality data to test them in a zero-shot setting.
>
> (2) There is a lack of challenging closed-ended tasks in the long context settings. Previous long document datasets were mostly open-ended tasks, e.g., summarization, but there are still fairness issues in evaluating LCLMs on these open-ended tasks, and the  GPT4-eval is costly.
>
> *Addressing the two issues:*
>
> To address the first problem, we utilized Claude and human labelers to clean the existing datasets and we paid much attention to data diversity.  For the second problem, we introduced three closed-ended tasks with low evaluation costs that are less susceptible to metric biases.
>
> *Detailed motivations for the 4 tasks annotated from scratch*
> 1. Coursera: We use this task to test the reasoning ability of LCLMs on lengthy and challenging courses. This dataset has multiple correct options and features big data and machine learning courses.  We also found that other models lagged behind GPT-4-32k by the largest margin on this task.
> 2. SFcition: In long-context scenarios, models are expected to answer questions based on contextual knowledge (input tokens) rather than the knowledge they learned during training[1]. Therefore, we annotated a scientific fiction dataset (True or False questions) where most of the answers to the questions *contradict* real-world knowledge.
> 3. LongFQA:  Finance is a crucial application scenario for long-context models, but current financial datasets, such as FinQA, are all short-context, which prompts us to annotate LongFQA. When labeling longFQA, we referred to some questions in FinQA and designed questions that require global information.
> 4. CodeU: Apart from finance, coding represents another scenario that lacks high-quality test data. We challenge the model to provide execution results for complex and lengthy code to test the model's understanding of long codes. The answers are labeled by running the code.
>
> More details for each dataset are in Appendix B. We will add more explanations to help readers better understand our paper.
>
> > **Q2**: The findings of the paper are intuitive.
>
> Although the serious issues with n-gram metrics have been revealed by previous research, the fact remains that almost all current long-context models still use ROUGE and F-1 as evaluation metrics for open-ended tasks. Many LCLMs claim to outperform commercial models on various tasks, but our metric experiments suggest this could be due to biased metrics. It's still challenging for open-source LCLMs to achieve a high win-rate compared to GPT3.5-16k on L-Eval.
>
> *Other interesting findings*
> * Reference-based metrics are very sensitive to length.  By simply informing the expected length, we improve Claude by **50** F-1 score on NQ.
> * We observe the performance disparity between LCLMs on open-ended and closed-ended tasks
> * The performance differences on retrieval tasks and reasoning tasks when increasing the RoPE base frequency.
>
> These findings are different from previous (or concurrent) works.
>
> > **Q3**: Shortcomings of current models and how L-Eval facilitates future research.
>
> *Shortcomings*:
>
> According to our experiments with L-Eval, the most significant issue with open-source LCLMs is that as the number of input tokens increases, the performance on open-ended tasks decreases. However, this issue is not easily noticeable from n-gram metrics where the ROUGE scores are still competitive. Our human eval shows one of the reasons is the weakened instruction-following ability.
>
> *How L-Eval contributes to the research of LCLMs*:
>
> 1. 	Our work can help researchers re-think previous methods.  It shows that scaled position embedding and further training enhance the model's usage of more input tokens (or information)  for closed-ended tasks. However, open-source LCLMs struggle with open-ended tasks, highlighting a need for better instruction understanding in tasks with varied question styles.
>
> 2. 	We are calling for the use of more fair evaluation metrics instead of solely relying on metrics like ROUGE, which rewards models simply for generating an output similar in length to the standard answer.
>
> 3. 	L-Eval integrates more closed-ended tasks, devoid of evaluation metric biases.  Notably, open-source LCLMs still significantly lag behind commercial models, and the low testing cost of closed-ended tasks can provide researchers with rapid and accurate feedback.
>
> **References**:
>
> [1] Neeman, Ella, et al. "Disentqa: Disentangling parametric and contextual knowledge with counterfactual question answering." arXiv preprint arXiv:2211.05655 (2022).

---

> > ### Author Response · Authors · 2023-11-21
> > **Looking forward to receiving your feedback**
> >
> > Dear Reviewer cNRy,
> >
> > In the previous section, we respond to your two main concerns about the motivations of data collection and insightful findings.  We would be very grateful if you could provide feedback on our rebuttal since the deadline is approaching soon. If there are any additional questions, please feel free to ask us. We will immediately respond to them.
> >
> > Best wishes,
> >
> > The Authors

---

### Official Review · Reviewer_xVYn · 2023-11-02

**Soundness:** 3 good
**Presentation:** 3 good
**Contribution:** 4 excellent
**Rating:** 8
**Confidence:** 4

**Summary:**

This paper proposes L-Eval, a new benchmark for evaluating long context language models (LCLMs).

The authors start by arguing that existing benchmarks have limitations in dataset construction and evaluation metrics for properly assessing LCLMs' capabilities. To address this, the authors construct L-Eval, which contains 20 diverse sub-tasks with over 500 long documents and 2000 human-labeled question-answer pairs. The tasks cover various domains, question styles, and input lengths up to 200k tokens and, importantly, contains both *closed-ended* (with exact evaluation) and *open-ended* tasks . To construct the benchmark, the authors annotate 4 datatsets from scratch, re-annotate 5 others, and aggregate other 12 datasets/tasks from previous literature, resulting in a total of 21 tasks.

The paper also studies limitations of traditional n-gram matching metrics for open-ended generation tasks. Experiments show these metrics often fail to correlate with human judgments, particularly when models’ outputs different significantly in length from the reference answer. The authors propose techniques like length-instruction (asking for answer of similar lenght to the reference) and using LLM-based evaluation to improve correlation.

Comprehensive experiments are conducted on L-Eval with 16 closed-access and open-source LLMs, and include a human-evaluation for the open-ended tasks.

Key findings include:

- Significant gaps remain between commercial and open-source LCLMs.
- While open-source LMs finetuned for long-context improve on closed-ended tasks, they struggle on open-ended tasks as input length increases, even compared to the non-long-context-finetuned counterparts.
- Scaled positional embeddings enhance retrieval but can hurt reasoning over long contexts.

**Strengths:**

- The problem tackled is very relevant: modelling long context is one of the most important open-problems in LM research, and one of the main difficulties has been lack of proper evaluation.
- The proposed benchmark is quite diverse, being composed of many domains and both closed-form and open-ended NLP tasks. It also seemed to have required a consirable amount of work given the amount of human-annotations involved.
- Very throughout analysis of the limitations of automatic metrics for open-ended tasks, including a potential easy fix (length instruction enhanced evaluation) to make them more reliable evaluators (albeit clearly human/model based eval still seems to be better).
- Throughout evaluation of current SotA models, both open-source and closed-access.

**Weaknesses:**

- My main criticism of the paper is due importance/relevance given to GPT-based evaluation. While I believe that this is probably better than lexical metrics, its unclear if it will lead to biased evaluations, particularly when evaluating GPT-based genarators. In my opinion, table 5 (human eval) is more important to include in the paper than Table 4, and should include results for human-eval of the GPT-4 model (so as to compare with the GPT-4 based eval).
- The paper has multiple typos and gramar errors. E.g.
    - “We inject come new”
    - “NTK-ware”
    - “directly exposure to LCLMs”
    - …
  I suggest the authors run the text by simple automatic grammar correction since I’m pretty sure it would catch most of these

**Questions:**

n/a

---

> ### Author Response · Authors · 2023-11-14
> **Responses to reviewer xVYn**
>
> Thank you for your insightful feedback. We summarize your main concerns as follows:
>
> >**Q1**: GPT-based evaluation is also biased and human eval is more important.
>
> Thank you for an interesting question. Due to the high cost of human evaluation for long-context tasks, we were only able to test seven models on 85 questions, which already required four days. The high cost prevents us from manually evaluating all the models. Therefore, we included GPT-4 evaluation results in the main paper. With the automatic evaluator, we were able to evaluate 21 different systems, providing a better understanding of current LCLMs. We agree that the GPT evaluators are also biased and could potentially favor GPT-generated content. However, in the GPT-4 evaluation, Claude-100k surpassed GPT-4-32k, achieving the best results. This may indicate that the bias is not as strong as presumed. We have modified the caption of Table 4 to highlight the potential bias of GPT-based evaluation and encourage readers to refer to Table 5 for more accurate results regarding open-ended tasks.
>
> > **Q2**: Grammar errors and typos:
>
> Thank you so much for pointing out the grammar errors. We have polished this paper and used some automatic grammar correction tools based on your suggestions.

---

### Official Review · Reviewer_jW5y · 2023-11-12

**Soundness:** 2 fair
**Presentation:** 2 fair
**Contribution:** 2 fair
**Rating:** 5
**Confidence:** 4

**Summary:**

The paper is focusing on the evaluation of long-context language models and proposes a new evaluation benchmark for the evaluation of long text generation that consists of 20 closed (reasoning and understanding) and open-ended (summarization) sub-tasks, 508 documents, 2000 human annotations for query-response pairs. The authors investigate the performance of existing evaluation metrics on this benchmark and find that n-gram matching metrics present poor correlation with human judgements, leading them to recommend the use of LLM-based evaluation metrics.

**Strengths:**

The paper is making an important contribution to the community which is in need of benchmarks for evaluating long-term text generation models. The authors present comprehensive statistics about the benchmark which includes both new and existing datasets. For the existing datasets, the authors put effort into cleaning and correcting misleading previous annotations. On the proposed benchmark, the authors evaluate the performance of existing reference-based evaluation metrics on both open-ended and close-ended tasks.

**Weaknesses:**

Abstract: “GPT-4 and Claude can largely preserve the reasoning ability in an extended context” - this statement is misleading, there is a significant body of work showing how the reasoning abilities of these models is limited particularly for long contexts / multi-hop reasoning tasks.

The paper  does not provide a clear and convincing argumentation on why the specific datasets they select are included in this benchmark. To what extent this was an informed decision made by the authors is unclear. In addition, there is no clear highlight of differences between the currently proposed benchmark and existing long sequence benchmarks in the literature (which the authors describe in Section 2.2). Why is L-eval superior to existing benchmarks is not clarified in this paper. As this paper has the potential to impact the research community, it should be motivated why researchers would use this benchmark instead of other existing long-context benchmarks.

It would be good to include length statistics for both inputs and references across the tasks in the benchmark, and maybe correlate this with the difficulty of solving a particular task. The paper only limits to briefly mentioning “The length of reference in L-Eval also varies significantly across tasks.”

The authors conduct evaluation of reference-based metrics on their proposed benchmark, and conclude that n-gram based metrics are inferior to LLM-based metrics for long-text generation. Furthermore, the main takeaway from the evaluation analysis is that authors recommend the use LLM-based metrics, however these come with problems that are not discussed in this paper. The blunt recommendation of LLM-based metrics without a thorough discussion of their biases and failure cases is a strong limitation of this paper.

“LLM evaluators have been reported to favor more detailed and lengthy answers”  - the authors do not discuss how the length statistics across the tasks in this benchmark have an impact on these evaluation biases. For the length-instruction based evaluation, accounting for the length of the reference answer may introduce additional biases: “We need a 50-word summary, where 50 is the number of words in the reference answer”.

Moreover, they claim that “Human evaluation This is the most accurate evaluation for open-ended tasks.” - While I agree with this statement, human evaluation is not flawless either. This paper does not present details of how the manual annotation was conducted, which does have an impact on the results reported and the conclusions of this paper.

**Questions:**

Contradictory information: In Introduction you mention “L-Eval has 20 sub-tasks, 4 sub-tasks are annotated from scratch (§3.1), 4 sub-tasks are re-annotated from the public datasets (§3.2), and the remaining 12 sub-tasks are manually cleaned from previous long sequence datasets.”, while in Section 3.2 you claim “We re-annotate 5 publicly available datasets in L-Eval.” How many publicly available datasets are you annotating?

Section 3.1. - Data Annotation from Scratch: for the 4 datasets presented, are these manually or automatically annotated? There is no information provided in this section.

“The average input length in L-Eval ranges from 4k to 60k”. Do you measure the length in tokens? If yes, this should be specified.

How is your benchmark different from from existing long sequences benchmarks?

Have you considered the behaviour of referenceless metrics on your benchmark?

---

> ### Author Response · Authors · 2023-11-14
> **Responses to reviewer jW5y**
>
> Thank you for your detailed review. Q1 – Q5 are our responses to your questions and W1 – W5 are our responses to the weaknesses you have mentioned.
>
> ---
>
> > **Q1**: How many datasets in L-Eval are re-annotated?
>
> We re-annotate **5** datasets: GSM-16shot, TopicRet, QuALITY, Openreview and SPACE. We appreciate this opportunity to clarify and have updated the Introduction section to reflect the correct information.
>
> >**Q2**: Are these 4 new datasets in L-Eval manually or automatically annotated?
>
> The new datasets in L-Eval are all manually annotated as we strongly oppose using GPT generated data for evaluation. The detailed annotation process is listed in Appendix B.
>
> >**Q3**: Do you measure the length in tokens?
>
> Due to the difficulty of measuring the length for code data,  L-Eval uses Llama2 tokenizer to calculate the length as the number of tokens for all datasets, and this is specified in the last line of Table 1 caption.
>
> > **Q4**: How is L-Eval different from existing long sequence benchmarks?
>
> To the best of our knowledge, the most related evaluation suites for testing long context language models include ZeroScrolls (Findings of EMNLP 2023)[1] and Longbench (Arxiv)[2]. In accordance with the ICLR 2024 review policy, these can be classified as concurrent work.
>
> *Differences*:
> * In contrast to previous work, all samples in L-Eval are manually selected. Furthermore, we filter out both unanswerable and mistaken questions to ensure high-quality data.
> * We refrain from relying on n-gram metrics for open-ended tasks, which leads to different conclusions.
> *  We annotate a larger number of closed-ended tasks, yielding results that are less likely to be influenced by metric biases.
> More details can be found in Section 2.2.
>
> > **Q5**: Referenceless metrics.
>
> Developing referenceless metrics for long-context language models remains an open question. It is still hard to develop a referenceless metric since LCLMs are in their early stage and the reference is considered an important aspect to evaluate outputs. So, L-Eval mainly relies on reference-based metrics and short-context LLMs as the judges.
>
> ---
>
> > **W1**: “GPT-4 and Claude can largely preserve the reasoning ability…” is misleading.
>
> This sentence is mainly based on the comparison between GPT-4/Claude and open-source models. We will add “compared with open-source models” at the end of this sentence.
>
> > **W2**: Why the specific datasets are selected in this benchmark
>
> Generally, we aim to construct a diverse and high-quality suite. When choosing specific datasets, it's logical to begin with publicly available long-sequence datasets, such as QMSum and NQ, due to their easy accessibility and popularity. Previous datasets are large-scale containing low-quality/unanswerable questions, so we manually check and correct the data. Noticing the metric bias in open-ended tasks, we find there is still a lack of closed-ended long sequences tasks motivating us to annotate 3 closed-ended tasks from scratch. To ensure domain diversity we also annotate the code and finance data.
>
> >**W3**: Include length statistics for both inputs and references across the tasks.
>
> The length statistics for input can be found in Table 1. We list the average reference length here:
>
> | Dataset       | Avg reference len |
> | ------------- | ------------------------ |
> | NQ            | 4                        |
> | narrativeQA   | 9                        |
> | Qasper        | 12                       |
> | multidoc2dial | 21                       |
> | QMSum         | 60                       |
> | LongFQA       | 61                       |
> | SPACE         | 97                       |
> | SummScreen    | 98                       |
> | bigpatent     | 124                      |
> | GovReport     | 261                      |
> | Openreview    | 273                      |
> | multi-news    | 274                      |
> We will add this information in Table 1.
>
>
> **References**
>
> [1] Shaham, Uri, et al. "ZeroSCROLLS: A Zero-Shot Benchmark for Long Text Understanding." arXiv preprint arXiv:2305.14196 (2023).
>
> [2] Bai, Yushi, et al. "LongBench: A Bilingual, Multitask Benchmark for Long Context Understanding." arXiv preprint arXiv:2308.14508 (2023).

---

> ### Author Response · Authors · 2023-11-14
> **Responses to reviewer jW5y (Part II)**
>
> > **W4**: The blunt recommendation of LLM-based metrics
> 1. *LLM judges have been widely used in modern open-ended benchmarks*
>
> Firstly, almost all modern open-ended benchmarks like AlpacaEval[3], MT-bench[4] Vicuna Bench[5] use LLM judges, discarding ROUGE and F-1. However, long context LLM still insists on using ROUGE or F-1. Compared with ROUGE and F-1, we highly recommend using LLM judges instead.  In long-context settings, we cannot provide the entire input to the judge; this forms a key difference. Consequently, we have designed a new judge prompt within this setting to better adapt to long-context situations.
>
> 2. *Our experiments have confirmed that the LLM judges can achieve better consistency with human evaluation.*
>
> We agree that automatic metrics for open-ended tasks including LLM judges have many drawbacks but we strongly disagree with the assertion that the recommendation of LLM-based judges is blunt. We have experimentally confirmed that LLM judges outperform n-gram metrics in the long-context setting by ranking generation results from seven models.  We adopt the widely used Kendall-Tau correlation coefficient [6][7] to measure the correlation between these automatic metrics and human assessments.  Specifically, we rank these model outputs (7 models x 85 questions) using various metrics (e.g., F1, ROUGE, human score, and GPT4) and compute the Kendall-Tau correlation coefficient with human judgment. Results indicate that the Kendall-Tau correlation coefficient of ROUGE is only **0.52**, while the GPT-4 judge achieves a significantly higher score of **0.95**, outperforming other metrics. Please refer to Appendix A2 for the details of the human evaluation setup.
>
> > **W5**: Potential biases and failure rate in LLM judges
>
> Thank you for your valuable advice. We have conducted a study on the biases and failure rate of LLM judges and will include these results in our camera-ready version. Results from the 85-question subset are shown in the following table. In this table, 'fail-rate' represents the cases where judgments from human judges and LLM judges are contradictory. 'b-first' represents the percentage of cases when a judge favors the first answer, and 'b-details' is the percentage of cases when a judge favors the more detailed answer.
>  |               | b-first | b-details | fail-rate | Error-samples |
> | ------------- | ------- | --------- | --------- | ------------- |
> | GPT-3.5 judge | 56%     | 68%       | 49/154    | 16            |
> | GPT-4 judge   | 31%     | 37%       | 22/170    | 0             |
>
>
> > **W6**: human evaluation is not flawless and the details of human evaluation.
>
> The human annotation process can be found in Appendix A2. All three annotators are PhD students. Annotating the 85 questions from 7 models (7x85 outputs) took approximately 4 days. Although we agree human evaluation is also biased, it is still generally believed to be more accurate than current automatic metrics.
>
>
> **References**
>
> [3]  Zheng, Lianmin, et al. "Judging LLM-as-a-judge with MT-Bench and Chatbot Arena." arXiv preprint arXiv:2306.05685 (2023).
>
> [4] Li, Xuechen, et al. "Alpacaeval: An automatic evaluator of instruction-following models." (2023).
>
> [5] Chiang, Cheng-Han, and Hung-yi Lee. "Can Large Language Models Be an Alternative to Human Evaluations?." arXiv preprint arXiv:2305.01937 (2023).
>
> [6] Zhang, Tianyi, et al. "Bertscore: Evaluating text generation with bert." arXiv preprint arXiv:1904.09675 (2019).
>
> [7] Ye, Seonghyeon, et al. "Flask: Fine-grained language model evaluation based on alignment skill sets." arXiv preprint arXiv:2307.10928 (2023)

---

> ### Author Response · Authors · 2023-11-19
> **Looking forward to receiving your feedback**
>
> Dear Reviewer jW5y,
>
> We have meticulously revised our manuscript in line with your suggestions and we think your primary concern pertains to the metrics. In response, we have now incorporated a preliminary examination of potential biases in LLM judges on our benchmark. Notably, while short-context benchmarks have primarily used LLM judges for open-ended tasks, LCLMs are still evaluated with n-gram metrics. L-Eval first demonstrates that LLM judges are also more accurate in long-context settings.
>
> Furthermore, compared to other studies that primarily focus on metrics, L-Eval's contributions are manifold: (1) dataset construction, (2) benchmarking 16 popular models, (3) presenting insightful findings on the current state of LCLMs.
>
> Thank you once again for your valuable feedback. We are wondering if there are any additional questions. Please feel free to discuss with us!
>
> Best regards,
>
> The Authors

---

### Public Comment · ~Jihyung_Youn1 · 2023-11-28
**If parsing is unavailable, it processes the texts as "A" in multiple-choice problem**

Thank you for your release; it has been really helpful to me.

However, as I saw you supplementary material, I stumbled upon something with which I cannot fully agree, especially regarding the evaluation logic in the multi-choice problem.


In both the ground truth and predicted output, the processing function for the multiple-choice task uses "A" when the clened result is shorter than 1.
(refer: process_gt_mc() and process_output_mc() from auto_eval.py in Evaluation directory)

It has been commented as a "random guess" but It does not operate as described in the comment.
Furthermore, if it operates as commented, the randomly guessed result will lead to an incorrect score.
If there is a question with the ground truth set as "A", It could be scored as a correct answer even if there is no actual answer, when it cannot parse the answer: because it will processed as "A".

Once again, thank you for your experiments and contribution with inference results, code, and data. I hope there will be corrections or comments to address the concerns I've raised.

---

### Meta-Review · Area_Chair_eHs6 · 2023-12-10

**Metareview:**

This paper proposes L-Eval, a benchmark of 20 tasks for evaluating how well LLMs utilize long-range context on both QA style tasks (e.g., multiple choice problems) and open-ended generation (e.g., summarization), where evaluation on the latter is done via GPT-4 (an "LLM judge"). Overall, all reviewers praise the motivation of the benchmark (more rigorous evaluations are definitely needed for long-context LLMs), the amount of work that went into its creation, and the comprehensive nature of the experiments. However, there were concerns about the selection of tasks and examples for the benchmark and how L-Eval differs from existing long-context benchmarks such as SCROLLS. Additionally, L-Eval relies on human annotation to both select examples from existing datasets and to validate the LLM judge; however, the human evaluation protocol is not well described or motivated (essentially a 1-5 Likert scale for each open-ended task) and conducted using PhD student annotators (what are their qualifications to judge long output texts? what is their interannotator agreement?). Finally, the authors do not address the potential of data contamination with L-Eval: what if the input documents are in the pretraining data of an LLM? what if the output labels are (or will be) in future pretraining/finetuning datasets? Overall, I think the paper can benefit from another round of revision.

**Justification For Why Not Higher Score:**

There are a couple major issues with this paper that lead me to recommending Reject instead of Accept (poster). First, there are already similar long-context benchmarks to L-Eval, and while the authors should be commended for cleaning/reannotating/filtering some of these existing datasets, there is still limited novelty in the work. Next, the human evaluation procedure seems questionable and was not adequately addressed in the response.

**Justification For Why Not Lower Score:**

N/A

---

### Decision · Program_Chairs · 2024-01-16

Reject